# TMetaNet: Topological Meta-Learning Framework for Dynamic Link Prediction

Hao Li [1]  Hao Wan [1]  Yuzhou Chen [2]  Dongsheng Ye [3]  Yulia Gel [4]  Hao Jiang [1]

## Abstract

Dynamic graphs evolve continuously, presenting challenges for traditional graph learning due to their changing structures and temporal dependencies. Recent advancements have shown potential in addressing these challenges by developing suitable meta-learning-based dynamic graph neural network models. However, most meta-learning approaches for dynamic graphs rely on fixed weight update parameters, neglecting the essential intrinsic complex high-order topological information of dynamically evolving graphs. We have designed Dowker Zigzag Persistence (DZP), an efficient and stable dynamic graph persistent homology representation method based on Dowker complex and zigzag persistence, to capture the high-order features of dynamic graphs. Armed with the DZP ideas, we propose TMetaNet, a new meta-learning parameter update model based on dynamic topological features. By utilizing the distances between high-order topological features, TMetaNet enables more effective adaptation across snapshots. Experiments on real-world datasets demonstrate TMetaNet's state-of-the-art performance and resilience to graph noise, illustrating its high potential for meta-learning and dynamic graph analysis. Our code is available at https://github.com/Lihaogx/TMetaNet.

## 1. Introduction

The complexity of dynamic graph evolution, characterized by continuous changes in nodes and edges, makes it difficult for existing graph learning algorithms to fully capture the temporal relationships and dynamics within these graphs (Feng et al., 2024). Traditional methods are often lim-

*Equal contribution  [1]Wuhan University, Wuhan, China [2]University of California, Riverside, USA [3]Hubei University of Automotive Technology, Shiyan, China [4]Virginia Tech, Blacksburg, USA. Correspondence to: Hao Jiang <jh@whu.edu.cn>.

*Proceedings of the $42^{nd}$ International Conference on Machine Learning*, Vancouver, Canada. PMLR 267, 2025. Copyright 2025 by the author(s).

ited by their training strategies and evaluation approaches, rendering them inadequate for handling the unpredictability and rapid shifts in graph structure (You et al., 2022). This has led to a growing interest in meta-learning, which provides a powerful framework for dynamic graph tasks by enabling models to quickly adapt to new changes with minimal data. Meta-learning-based approaches have demonstrated state-of-the-art performance in downstream tasks, such as dynamic link prediction (You et al., 2022; Yang et al., 2022; Zhu et al., 2023b).

The core idea of meta-learning lies in updating the model parameters across different tasks to enable adaptation to new tasks (Gharoun et al., 2024; Vettoruzzo et al., 2024). For a discrete-time dynamic graph $\mathcal{G}_T = \{G_1, G_2, \ldots, G_T\}$, ROLAND (You et al., 2022) introduced a novel training strategy called live update, which treats the task of predicting new links from $G_t$ to $G_{t+1}$ as a separate task. This approach helps avoid the large memory consumption typically associated with such tasks. Building on this, WinGNN (Zhu et al., 2023b) removes explicit time encoding and incorporates temporal information into the graph neural network model through the self-adaptive aggregation of model parameters across windows. Figures 1 (a) and (b) illustrate the main training processes of ROLAND and WinGNN.

However, dynamic graph snapshots are not merely separate tasks; they form part of a larger, continuously evolving system. In this context, the way model parameters change across tasks plays a crucial role in determining the success of meta-learning. The aforementioned methods only account for the impact of time on parameter changes, neglecting the unique evolution of structures in dynamic graphs. Inspired by task-aware meta-learning (Chen & Zhang, 2021; Peng & Pan, 2023), it is crucial to find a representation of dynamic graphs that can accurately measure the differences between tasks and guide the parameter update process.

Naturally, our attention turns to persistent homology (PH), a tool from computational topology used to extract and learn descriptors of the data's "shape" (Carlsson & Gabrielsson, 2020; Pun et al., 2022). The persistent homology method constructs topological features of a graph's multi-scale information, capturing both low-order and high-order details. In recent years, PH has been widely combined with graph neural networks (GNNs) and applied to various down-

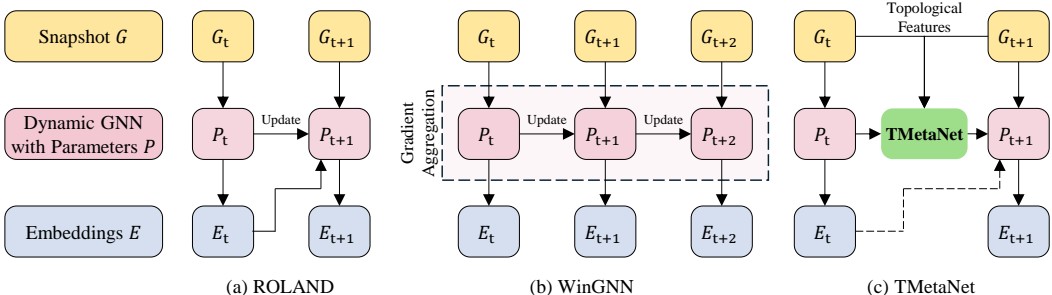

*Figure 1.* The parameter update mechanisms based on meta-learning in ROLAND, WinGNN, and TMetaNet. (a) In ROLAND, model parameters between adjacent time slices are updated using fixed meta-learning weights. (b) WinGNN performs parameter updates between adjacent time slices with a fixed learning rate, incorporating a window gradient aggregation mechanism to replace explicit time encoding, making the embedding $E_t$ invisible to the model $P_{t+1}$. (c) TMetaNet learns the learning rate for dynamic GNN model parameters using high-order topological features from adjacent time slices. The dashed line represents the training method that simultaneously adapts ROLAND and WinGNN.

stream tasks such as graph classification, node classification, anomaly detection, and link prediction (Yan et al., 2021; Chen et al., 2021a; Li et al., 2024; Immonen et al., 2023; Kerber & Russold, 2024; Ye et al., 2025). The ability of PH to capture high-order information in a graph reflects the inherent structural differences between snapshots and naturally complements meta-learning-based GNNs (Chen et al., 2022c). Our goal is to bridge the emerging concepts of PH and meta-learning for time-evolving graphs, using topological features to enhance the parameter update mechanism of dynamic GNNs and to further boost their expressive power. This brings two key challenges: (i) **How to effectively generate stable persistent homology features for dynamic graphs**? and (ii) **How to efficiently utilize persistent homology features to guide the parameter updates of dynamic GNN models**?

To address the above challenges, we propose the Dowker Zigzag Persistence (DZP), a computationally efficient and stable dynamic graph persistent homology representation method based on Dowker complex and zigzag persistence. DZP effectively captures high-order structural information of dynamic graphs, using only a small subset of nodes, but without sacrificing the accuracy. Based on DZP, we design a new topological meta-learning framework TMetaNet, which updates the model parameters by leveraging the high-order features of the tasks as shown in Figure 1 (c). By incorporating DZP into the meta-learning framework, it not only improves the model's ability to capture deep structural differences between graph snapshots but also enhances the adaptability of dynamic GNNs to evolving graph topologies. Our main contributions are summarized as follows:

1. We propose a new Dowker Zigzag Persistence (DZP) approach which offers a computationally efficient and robust representation of the key topological properties of dynamic graphs, by harnessing the strengths of the Dowker complex and zigzag persistence.

2. We derive theoretical stability guarantees of the resulting DZP representations and show their utilities to yield a comprehensive description of the topological structure of discrete-time dynamic graphs, enabling a deeper understanding of the graph evolution and its underlying patterns over time.

3. Armed with DZP, we introduce the notion of topological meta-learning on dynamic graphs and develop a new mechanism for parameter updates that integrates topological feature enhancement. The proposed TMetaNet framework allows the model to more effectively and accurately capture the evolving topology of dynamic graphs under a broad range of uncertainties.

4. We illustrate the utility of TMetaNet in application to link prediction in directed and undirected dynamic graphs. Our extensive experiments indicate that TMetaNet outperforms state-of-the-art benchmarks up to 74.70% and yields up to 31.1% gains in robustness.

## 2. Related Work

**Meta-learning for Dynamic GNNs** has become one of the hottest research topics in recent years due to its potential to adapt the resulting model to new tasks with minimal retraining in a dynamic environment. In particular, meta-learning extracts prior knowledge from training tasks to apply to new tasks with limited data (Gharoun et al., 2024). In turn, model-agnostic meta-learning (MAML) is a meta-learning framework which is popular due to its versatility in a broad range of tasks and scenarios (Finn et al., 2017). In static graphs, it is used with GNNs for few-shot predictions (Zhou et al., 2019; Bose et al., 2019; Chauhan et al., 2020; Yao et al., 2020). For dynamic graphs in spirit of MAML,

MetaDyGNN (Yang et al., 2022) extracts multi-level knowledge for temporal link prediction, while MetaRT (Zhu et al., 2023a) predicts time-aware knowledge triples. In discrete-time dynamic graphs, ROLAND (You et al., 2022) uses live update training to model tasks as predictions between snapshots, maintaining adaptability with fixed-weight updates. WinGNN (Zhu et al., 2023b) predicts links without explicit time encoders using window gradient aggregation. TMetaNet focuses on the more common discrete-time dynamic graphs, combining perspectives from meta-learning and graph neural networks to explore meta-learning parameter update mechanisms enhanced by persistent homology features, bringing new insights to the field.

**Zigzag Persistence** is a tool from computational topology, allowing us for more effective and reliable analysis of temporal evolution processes (Carlsson & Silva, 2010; Tausz & Carlsson, 2011; Carlsson et al., 2019). It has shown promising results in a range of downstream tasks, such as neuroscience (Mata et al., 2015; Chowdhury et al., 2018), swarming phenomena in biology (Kim et al., 2020), cybersecurity (Myers et al., 2023a), and power distribution planning (Chen et al., 2023). However, the application of zigzag persistence in deep learning (DL) remains largely unexplored. Some studies have transformed zigzag persistence into topological representations directly usable in DL, such as zigzag persistence image, zigzag filtration curves, and zigzag spaghetti (Chen et al., 2021b; 2022b;a; Chen & Gel, 2025), as well as explored combination of zigzag with multi-persistence (Coskunuzer et al., 2024). However, zigzag persistence is limited due to its scalability restrictions and the associated high computational costs (Dey & Hou, 2021; 2024). To further apply zigzag persistence to the representation of dynamic graphs, new approaches based on weaker complexes such as Dowker complex which effectively reduce computational complexity (Choi et al., 2024; Li et al., 2024), constitute a promising research direction. To achieve higher scalability, in this paper, we propose TMetaNet which bridges the gap between meta-learning and persistent homology. Through the efficient and stable dynamic graph topological representation learning based on Dowker Zigzag Persistence (DZP), we provide a new parameter update mechanism for dynamic graph meta-learning. This provides new insights into the link prediction problem in large-scale dynamic graphs.

## 3. Background

Persistent homology is a tool for analyzing multi-scale topological structures in data. In the context of graph theory, persistent homology constructs a sequence of nested topological spaces (such as simplicial complexes) and studies the evolution of their topological features (like connected components, cycles, etc.) as the scale parameter varies.

(Carlsson & Gabrielsson, 2020; Chazal & Michel, 2021; Hensel et al., 2021) (for more details on the concept of persistent homology on graphs, see Appendix A.)

Leveraging the mathematical the theory of quiver representations, zigzag persistence (ZP) is a generalization of the conventional tools of persistent homology (PH), allowing, in contrast to PH, to consider topological spaces which are connected via inclusions going into multiple directions (Carlsson & Silva, 2010; Tausz & Carlsson, 2011). As such, ZP is particularly well suited to study topological properties of dynamic objects and has recently gained increasing attention in various data analytics tasks involving time-varying processes (Chen et al., 2021b; 2022a; Myers et al., 2023b; Ma et al., 2025). In particular, for a discrete-time dynamic graph $\mathcal{G}_T = \{G_1, \ldots, G_T\}$, the zigzag filtration over graph snapshots takes the following form:

$$G_1 \hookrightarrow G_1 \cup G_2 \hookleftarrow G_2 \hookrightarrow G_2 \cup G_3 \hookleftarrow G_3 \cdots \quad (1)$$

$$G_k \cup G_{k+1} = (V_k \cup V_{k+1}, E_k \cup E_{k+1}). \quad (2)$$

where each morphism $\hookrightarrow$ represents the inclusion of one graph into the union of itself with the next graph, while $\hookleftarrow$ denotes the reverse inclusion. This zigzag sequence allows for both the addition and potential removal of nodes and edges as the graph evolves. Based on the zigzag sequence of graphs, with a fixed scale parameter $\xi$, we obtain the corresponding zigzag complex sequence:

$$
\begin{array}{ccccc}
C(G_1, \xi) & & C(G_2, \xi) & & C(G_3, \xi) \quad \cdots \\
& \searrow & \swarrow & \searrow & \swarrow \\
& C(G_1 \cup G_2, \xi) & & C(G_2 \cup G_3, \xi)
\end{array}
,$$

Armed with this zigzag construction, we can track the evolution of various topological features.

**Definition 3.1** (Zigzag Persistence Diagram). Given a zigzag filtration $\mathcal{F} = \{G_t \leftrightarrow G_{t\pm 1}\}_{t=1}^T$ and filtration scale parameter $\xi$, the $k$-dimensional Zigzag Persistence Diagram $Dgm_k(\mathcal{F})$ is a multiset of points in the extended plane $\overline{\mathbb{R}}^2$ where:

$$Dgm_k(\mathcal{F}) = \{(b_i, d_i) \in \mathbb{R}^2 \mid b_i < d_i\} \cup \Delta. \quad (3)$$

For the definition of dimension k, see A.2. Each point $(b_i, d_i)$ corresponds to a $k$-dimensional topological feature that is born in the complex $C(G_{b_i}, \xi)$ and dies in $C(G_{d_i}, \xi)$ at scale $\xi$. If the feature appears in $C(G_{b_i} \cup G_{b_i+1}, \xi)$ or disappears in $C(G_{d_i} \cup G_{d_i+1}, \xi)$, the corresponding coordinate is $(b_i + 1/2)$ or $(d_i + 1/2)$, respectively.

These birth and death times are represented as points in a Zigzag Persistence Diagram, which visualizes the persistence of topological features across different snapshots.

By converting the zigzag persistence diagram into a metric space, such as the zigzag persistence image (Chen et al., 2021b), the topological structure information of dynamic graphs can be applied to downstream machine learning tasks. Despite being a promising tool for studying a broad range of time varying objects, ZP remains substantially underexplored in practice due to its often prohibitive computational costs.

## 4. Dowker Zigzag Persistence

Although the inherent nature of zigzag persistence (ZP) appears as a perfect fit to capture the topological characteristics of dynamic graph evolution, computational complexity remains the major roadblock on the way of wider adoption of ZP in practice. For example, for 0- and 1-dimensional features on graphs, the currently best available algorithm to compute ZP has complexity of $O(m \log^2(N) + m \log(m))$ and $O(m \log^4(n))$ respectively, where $m$ is the length of the filtration and $N$ is the number of nodes used to construct the complex in the graph (Dey & Hou, 2021; 2024).

One intuitive idea to address this fundamental problem is to use somehow only a subset of the available nodes, when computing ZP. *However, can we do so, without sacrificing the topological information?* Fortunately, the answer to this question is positive if we invoke the notion of a Dowker complex on graphs. Dowker complex belongs to the family of weaker complexes which also includes, for example, witness complex (Ghrist, 2014; Chazal et al., 2014; Chowdhury & Mémoli, 2018), and also may be viewed as the witness complex counterpart on graphs (for more discussion on similarities and differences of witness and Dowker complexes see (Aksoy et al., 2023)). Dowker complex has been recently successfully applied to such tasks as adversarial graph learning (Arafat et al., 2025), GNN pre-trainng (Liang et al., 2025), and dynamic link prediction (Li et al., 2024). (Note that there are two forms of the Dowker complexes, and the Dowker complex considered in this paper is not the same as the Dowker complex studied by (Liu et al., 2022).)

The ultimate idea is to assess shape of the graph based only on a substantially smaller subset of nodes, called *landmarks*, while using all other remaining nodes as *witnesses* which dictate appearances of simplices in the Dowker complex.

**Definition 4.1. Dowker Complex.** For a snapshot $G_t = (V_t, E_t)$, the Dowker Complex $D(G_t) = D(L_t, W_t)$ is a simplicial complex constructed from the landmark set $L_t$ and the witness set $W_t$ as follows:

$$D(G_t) = \{\sigma \subseteq L_t \mid \exists w \in W_t \text{ such that} $$
$$\forall l \in \sigma, \ d(l, w) \leq \delta\},$$

where $\varepsilon$ represents the maximum allowable distance between landmark nodes and witness nodes. Each simplex $\sigma$

in $D(L_t, W_t)$ corresponds to a subset of landmark nodes. A $k$-simplex is included in $D(L_t, W_t)$ if there exists at least one witness node $w \in W_t$ that is within distance $\varepsilon$ from every landmark node in $\sigma$.

To select the suitable set $L_t$ of landmark nodes for the Dowker complex, we leverage the $\varepsilon$-nets algorithm (De Silva & Carlsson, 2004), which ensures computational efficiency without loss of topological information, thereby addressing the key roadblocks inherent for applications of ZP in dynamic graphs.

**Definition 4.2. $\varepsilon$-Net.** An $\varepsilon$-net $L_t \subseteq V_t$ satisfies $\forall v \in V_t, \ \exists l \in L_t$ such that $d(v, l) \leq \varepsilon$. Additionally, the $\varepsilon$-net is maximal, meaning that for any two distinct landmark nodes $l_i, l_j \in L_t$, the distance between them exceeds $\varepsilon$: $\forall l_i, l_j \in L_t, \ l_i \neq l_j \Rightarrow d(l_i, l_j) > \varepsilon$. (See Appendix C.1 for the pseudocode of the $\varepsilon$-nets algorithm.)

As a fundamental concept in computational topology, the $\varepsilon$-net ensures that $L_t$ is a representative subset of nodes and accurately captures graph topology without unnecessary redundancy (De Silva & Carlsson, 2004). Once the $\varepsilon$-net establishes the set $L_t$ of landmarks, the remaining nodes in $V_t$ are designated as witnesses $W_t = V_t \setminus L_t$. This partition facilitates the subsequent construction of the Dowker complex by establishing a bipartite relationship between landmark and witness nodes based on their mutual distances. (For more discussion of the $\varepsilon$-net properties, please see Arafat et al. (2019)). Armed with this construction, we now turn to equip zigzag filtration with the Dowker complex to capture the most essential topological information in dynamic graphs.

**Definition 4.3. Dowker Zigzag Persistence (DZP).** For a discrete-time dynamic graph sequence $\mathcal{G}_T = \{G_t\}_{t=1}^{T}$ with $G_t = (V_t, E_t)$, based on Eq. 1, the evolution of the graph is captured through the bidirectional sequence of Dowker complexes:

$$D(G_1, \varepsilon, \delta) \hookrightarrow D(G_1 \cup G_2, \varepsilon, \delta) \hookleftarrow D(G_2, \varepsilon, \delta) \hookrightarrow \cdots$$

Note that the Dowker Zigzag Persistence (DZP) does not require landmarks to remain fixed across different time snapshots. The varying landmarks across snapshots are naturally accommodated by zigzag persistence' interleaving mechanism. Through the canonical inclusion maps $D(G_t, \varepsilon, \delta) \hookrightarrow D(G_t \cup G_{t+1}, \varepsilon, \delta) \hookleftarrow D(G_{t+1}, \varepsilon, \delta)$, topological features persist when their corresponding simplices exist in both adjacent complexes or the intermediate union complex. This ensures consistency in recording topological features despite landmark variations.

Furthermore, since $\varepsilon$-net cardinality is bounded by $n' = n/(\varepsilon+1)+2n/d$, where $d$ is the diameter of the graph (Choi et al., 2024), the DZP reduces the complexity of computing 0- and 1-dimensional features on dynamic graphs

to $O(m \log^2(n') + m \log(m))$ and $O(m \log^4(n'))$ respectively, which demonstrates substantial improvement in computational costs.

We now show that DZP enjoys the important stability guarantees, ensuring robustness of DZP to uncertainties. To prove the stability of DZP, inspired by Ye et al. (2023), we first introduce the structural metric $d_{\mathcal{G}}$ based on $\epsilon$-interleaved dynamic graphs which is defined as follows:

**Definition 4.4. Structural Metric of Graph Discrepancy.**
For two discrete-time dynamic graphs $\mathcal{G}^X$ and $\mathcal{G}^Y$, if they satisfy Definition B.7 of $\epsilon$-interleaved dynamic graphs, the structural metric of discrepancy between $\mathcal{G}^X$ and $\mathcal{G}^Y$ is given by:

$$d_{\mathcal{G}}(\mathcal{G}^X, \mathcal{G}^Y) := \frac{1}{2} \inf_{R_1, R_2} \max \begin{cases} \text{dis}_V(R_1) + \text{dis}_V(R_2), \\ \text{dis}_\omega(R_1) + \text{dis}_\omega(R_2), \\ \max_t C_t(R_1, R_2) \end{cases}$$

The structural metric $d_{\mathcal{G}}$ quantifies dynamic graph differences through node coverage discrepancy ($\text{dis}_V$), weight divergence ($\text{dis}_\omega$), and bidirectional co-distortion analysis. (For a more detailed discusson of $d_{\mathcal{G}}(\mathcal{G}^X, \mathcal{G}^Y)$ see Appendix B.8.)

**Theorem 4.5. Dowker Zigzag Persistence Stability.** *For $\epsilon$-interleaved dynamic graphs $\mathcal{G}^X, \mathcal{G}^Y$ with structural metric $d_{\mathcal{G}}(\mathcal{G}^X, \mathcal{G}^Y)$, with their respective Landmark sets $\mathcal{L}^X$ and $\mathcal{L}^Y$, witness sets $\mathcal{W}^X$ and $\mathcal{W}^Y$ given according to Definition 4.2, their Dowker Zigzag Persistence diagrams satisfy:*

$$d_B \left( \text{Dgm}_{DZP}(\mathcal{G}^X), \text{Dgm}_{DZP}(\mathcal{G}^Y) \right) \leq K \cdot d_{\mathcal{G}}(\mathcal{G}^X, \mathcal{G}^Y)$$

*where constant $K$ depends only on the homological dimension $k$.*

The proof of the theorem is given in Appendix B. This stability result ensures that the smaller differences among graphs tend to result in smaller differences among the zigzag persistence diagrams associated with DZP. The theorem provides theoretical foundations for stable meta-learning parameter updates based on higher-order graph structures, which are beneficial for more effective and robust dynamic link prediction. This unified framework ensures topological stability while enabling noise-resilient meta-learning updates. In particular, our experiments in 6.4 illustrate the robustness of DZP to noise, while the results in 6.3 indicate consistency of the DZP conclusions with respect to varying sizes of landmarks and witnesses on downstream tasks.

## 5. Topological Meta-Learning on Dynamic Graphs

In this section, we introduce the proposed TMetaNet, designed to enhance link prediction in dynamic graph neural networks. As illustrated in Figure 2, TMetaNet consists of a Topological Signature Generator and a Topological Learning Rate Adaptor. In the Topological Signature Generator, we compute the Dowker Zigzag Persistence diagrams and the corresponding persistence images for discrete-time dynamic graphs. In the Topological Learning Rate Adaptor, we employ convolutional neural networks to extract topological feature differences between adjacent time snapshots, and use fully connected layers to learn a scalar as the learning rate. Finally, we apply this learning rate to the weight updates of the dynamic graph neural network, thereby achieving a topology-enhanced meta-learning parameter update process which results in a stable topology-enhanced meta-learning-based dynamic graph neural network.

### 5.1. Topological Signature Generator

Topological Signature Generator is used to compute the Dowker Zigzag Persistence diagrams $Dgm_{DZP}$ and the corresponding persistent image $ZPI_{DZP}$ (here we abbreviate them as $Dgm$ and $ZPI$ respectively) for discrete-time dynamic graphs. For a snapshot $G_t$, we first compute the Dowker complex $D_n(G_t)$ under parameters $\varepsilon_n$ and $\delta_n$.

$$D_n(G_t) = \text{Dowker}(L_{t,n}, W_{t,n}, \delta_n),$$
$$\{L_{t,n}, W_{t,n}\} = \varepsilon\text{-}seed(\{G_{t-w}, \ldots, G_t\}, \varepsilon_n).$$

Here, $\varepsilon_n$ and $\delta_n$ are parameters for computing the $\varepsilon$-net and Dowker complex $D_n(G_t)$, respectively. $\varepsilon$-seed is the seed-based $\varepsilon$-net generation algorithm. The input to $\varepsilon$-seed is a dynamic graph sequence with a fixed window length $w$. Through the expansion of the seed node set, it generates the $\varepsilon$-net corresponding to $G_t$. $\varepsilon$-seed ensures the consistency of the $\varepsilon$-net expansion, making the landmark node sets of adjacent time snapshots consistent. Algorithm details are provided in Appendix C.1. The construction of the Dowker complex $D_n(G_t)$ is shown in Definition 4.1.

After obtaining Dowker complexes for each snapshot, for the snapshot sequence $\mathcal{G}_t = \{G_{t-w}, \ldots, G_t\}$ and its corresponding Dowker complex sequence $\mathcal{D}_n(\mathcal{G}_t) = \{D_n(G_{t-w}), \ldots, D_n(G_t)\}$, we compute the zigzag persistence diagram for each Dowker complex sequence

$$Dgm_{n,k}(\mathcal{G}_t) = \mathbf{\Xi}_k(\mathcal{D}_n(G_t)),$$

where $\mathbf{\Xi}$ denotes the function that computes zigzag persistence. By constructing the sequence as shown in Eq. 1, we compute the $k$-dimensional zigzag persistence diagram. Each persistence diagram is a multiset of two-dimensional points $(x, y)$, representing the birth and death times of $k$-dimensional topological features. For each persistence diagram $Dgm_{n,k}(\mathcal{G}_t)$, we compute its ZPI $Z_{n,k}$, where $Z_{n,k}$ represents the $k$-dimensional ZPI under parameters $\varepsilon_n$ and $\delta_n$. See Appendix C.2 for more calculation details.

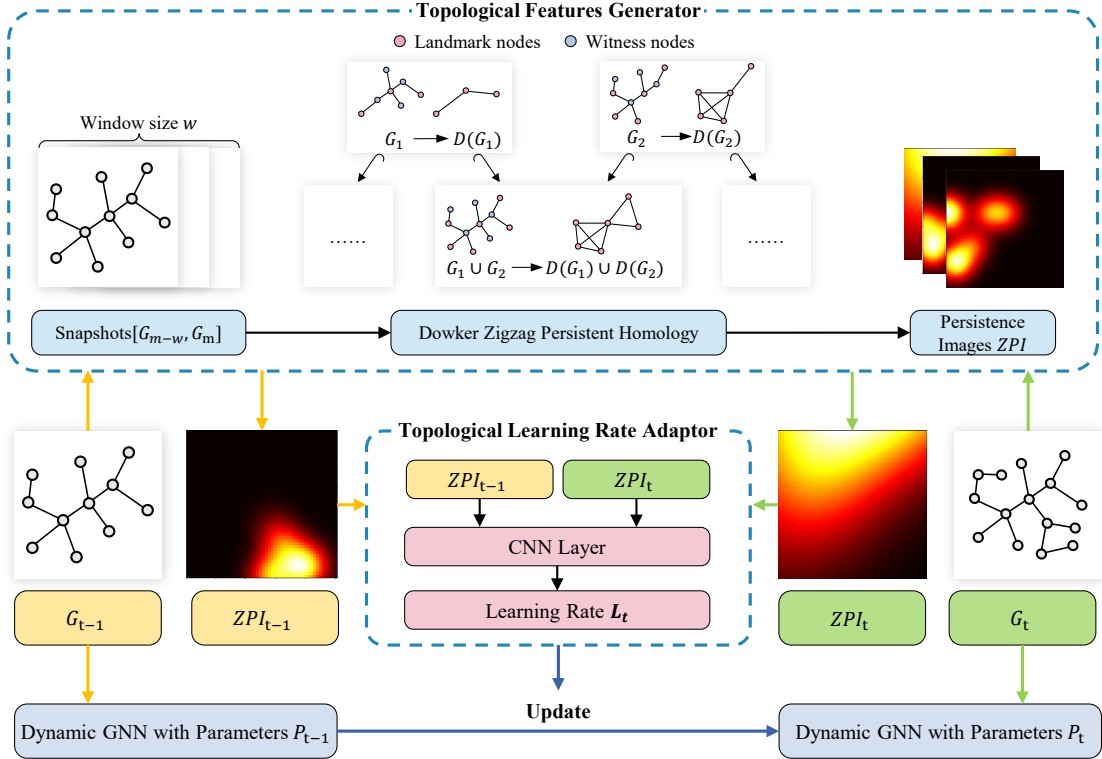

Figure 2. The overview of TMetaNet.

## 5.2. Topological Learning Rate Adaptor

For adjacent snapshots, e.g., $G_t$ and $G_{t+1}$, the Topological Signature Generator computes their corresponding ZPI sequences $Z_{n,k}(\mathcal{G}_t)$ and $Z_{n,k}(\mathcal{G}_{t+1})$. Subsequently, the Topological Learning Rate Adaptor extracts the difference between adjacent ZPIs through a convolutional neural network, and uses this as input to learn a scalar through fully connected layers, which serves as the learning rate for the dynamic GNN model. For adjacent ZPI sequences $Z_{n,k}(\mathcal{G}_t)$ and $Z_{n,k}(\mathcal{G}_{t+1})$, we then compute and obtain their difference through a convolutional neural network (CNN) and apply it as input to learn a scalar through fully connected layers, which serves as the learning rate for the dynamic GNN model. The specific formulas are as follows:

$$\Delta_t Z_{n,k} = Z_{n,k}(\mathcal{G}_{t+1}) - Z_{n,k}(\mathcal{G}_t),$$
$$r = \text{FC}(\sigma(\text{CNN}(\Delta Z_{n,k}))),$$

where $\Delta Z_{n,k}$ represents the difference between adjacent ZPI sequences, CNN represents the convolutional neural network layer (in TMetaNet, we use a simple CNN network with residual connections), $\sigma$ is the non-linear activation function, FC represents the fully connected layer, and $r$ represents the learned scalar learning rate.

After obtaining the learning rate, we apply it to the weight updates of the dynamic graph neural network, resulting in a topology-enhanced meta-learning parameter update process.

$$w_{t+1} = w_t - \eta \cdot r \cdot \nabla_w \mathcal{L}(w_t), \tag{4}$$

where $w_t$ represents the weights of the dynamic graph neural network at time $t$, $r$ is the scalar learned through the topological learning rate adaptor, and $\nabla_w \mathcal{L}(w_t)$ is the gradient of the loss function $\mathcal{L}$ with respect to weights $w_t$.

## 5.3. Training and Evaluation

For dynamic graph representation learning and use cases, ROLAND and WinGNN propose two different training strategies. ROLAND's live update training strategy divides each snapshot into training, validation, and test sets, and evaluates the model on each time slice. WinGNN, on the other hand, arranges snapshots in chronological order, with the first 70% as the training set and the last 30% as the test set. From a meta-learning perspective, ROLAND's training, validation, and test tasks belong to the same distribution, while WinGNN's training and test tasks belong to different distributions. ROLAND focuses more on model evaluation across the entire time scale, while WinGNN emphasizes prediction for a future time period. Appendix C.3 provides an illustration of different task splitting strategies. Compared to ROLAND and WinGNN, TMetaNet focuses on model parameter updates between adjacent time snapshots and

can be applied to both training and evaluation approaches mentioned above, as well as their corresponding metrics.

# 6. Experiments

**Datasets and Baselines**  We conduct experiments on six public datasets, which are widely used benchmarks for evaluating the performance of dynamic link prediction, i.e., (1) Bitcoin-OTC (OTC) and Bitcoin-Alpha (Alpha) are trust networks from transactions on different Bitcoin platforms (Kumar et al., 2018b; 2016). (2) Reddit-Body (Body) and Reddit-Title (Title) datasets are from the REDDIT platform, representing hyperlink networks in post titles and bodies, respectively (Kumar et al., 2018a). (3) UCI-Message (UCI) consists of private messages between users (Panzarasa et al., 2009). (4) ETH-Yocoin is derived from the Yocoin transaction network on Ethereum blocks (Li et al., 2020). Detailed dataset information can be found in the Appendix D.1. We compare TMetaNet with the following baseline methods: (1) GCN-L and GCN-G (Manessi et al., 2020). (2) EvolveGCN (Pareja et al., 2020). (3) WinGNN (Zhu et al., 2023b). (4) ROLAND (You et al., 2022). (5) DeGNN (Zheng et al., 2023). Details of these methods are provided in the Appendix D.2. The experimental settings can be found in Appendix D.3.

## 6.1. Link Prediction Performance

In this section, we present the performance of TMetaNet on dynamic link prediction tasks under different settings. The live update setting refers to the training strategy used in ROLAND, while the WinGNN setting refers to the training strategy used in WinGNN. Evaluation metrics include accuracy (ACC) and mean reciprocal rank (MRR), AUC, Recall@1, Recall@3, and Recall@10. Specific training settings are also provided in the Appendix D.3. For the analysis of running time, please refer to Appendix D.4. The results for accuracy and MRR metrics are shown in Table 1. The proposed TMetaNet demonstrates consistent superiority across diverse experimental settings and datasets. Our topological meta-learning framework achieves statistically significant improvements ($p < 0.05$) in different settings. On average, TMetaNet achieves 3.29% ACC improvement and 5.52% MRR improvement in the ROLAND setting, while showing even more pronounced performance in the WinGNN setting with 3.27% and 24.08% improvements respectively. This enhanced performance in the WinGNN setting may be attributed to the model's task of predicting future links, where TMetaNet's dynamic learning rate adaptation mechanism demonstrates better adaptability to such prediction tasks. These results collectively validate that topological persistence features provide effective structural signatures for meta-learning in dynamic graph scenarios. The performance gap is especially evident in datasets with

more dynamics, where topological signals are stronger and better leveraged by TMetaNet, while in more homogeneous cases such as Reddit-Body under the ROLAND setting, the gains are less significant. Other metrics are presented in Appendix D.5.

On ALPHA data, compared to VR complex-based zigzag persistence, our method reduces the computational overhead by 46% on average per snapshot during complex construction. When dealing with extremely large-scale graphs, we can sample from snapshots or remove nodes according to degree centrality from low to high to obtain subgraphs that preserve global higher-order features, and then calculate the learning rates.

## 6.2. Ablation Studies

In this section, we conduct ablation experiments on the Bitcoin-Alpha and UCI-Message datasets to verify the effectiveness of the model parameter update process guided by topological features in TMetaNet. We use three different parameter update settings. (1) Topo: Referring to the parameter update process based on DZP images. (2) Random: Replacing DZP images with a random image of the same dimensions as the original image. (3) Dist: Replacing each DZP image with an image having the same statistical characteristics as the original image. The experimental results are shown in the Table 2.

The experimental results demonstrate that the Topo setting, using the zigzag persistence diagram, consistently outperforms both Random and Dist across all metrics. Specifically, Topo achieves higher accuracy and MRR, showing the clear advantage of leveraging topological features over random replacements. Furthermore, while Topo outperforms Dist in both accuracy and MRR, the improvement is more pronounced when compared to the Random setting. This indicates that vectors following the same distribution also bring some performance improvement.

## 6.3. Sensitivity Analysis

In this section, we conduct experiments using the UCI-Message dataset to analyze the sensitivity of the DZP parameters. The specific parameters include the combination of parameters $\varepsilon$ and $\delta$ which are used to extract the Dowker complex when constructing higher-order features, and the window length $w$ which is used to construct the zigzag persistence.

As shown in Figure 3, each line represents a parameter combination $[(\varepsilon, \delta)]$ with window length $w$ on the horizontal axis. In the ROLAND setting, performance drops significantly as the window size increases. The $(2, 2)$ configuration achieves the highest MRR when $w = 5$, but shows a sharp decline when using the full-window setup, suggesting

*Table 1.* Dynamic link prediction performance of different models under different experimental settings. The best results are highlighted in bold, and the second-best results are underlined. Each experiment is repeated five times, and the mean and standard deviation are reported(omitting %). ∗ indicates statistical significance ($p$-value $< 0.05$) between TMetaNet and the second-best result.

| | | | | | LIVE UPDATE SETTING (ROLAND) | | | | |
|---|---|---|---|---|---|---|---|---|---|
| DATASET | METRIC | GCN-L | GCN-G | EGC-O | EGC-H | ROLAND | DEGNN | TMETANET | IMPR. |
| ALPHA | ACC | $73.81 \pm 0.72$ | $66.90 \pm 1.12$ | $71.34 \pm 1.89$ | $70.86 \pm 1.78$ | $83.18 \pm 2.77$ | $76.7 \pm 1.33$ | $*86.84 \pm 1.02$ | 8.14% |
| | MRR | $9.87 \pm 0.45$ | $9.00 \pm 1.32$ | $5.27 \pm 0.47$ | $1.87 \pm 1.42$ | $15.23 \pm 0.47$ | $12.5 \pm 1.03$ | $17.68 \pm 0.55$ | 11.8% |
| OTC | ACC | $77.65 \pm 1.11$ | $70.98 \pm 2.98$ | $72.33 \pm 1.09$ | $66.58 \pm 1.76$ | $84.77 \pm 1.06$ | $76.9 \pm 1.33$ | $*85.89 \pm 1.22$ | 1.32% |
| | MRR | $17.13 \pm 0.72$ | $16.98 \pm 1.32$ | $10.27 \pm 0.47$ | $6.87 \pm 1.42$ | $17.59 \pm 0.75$ | $15.0 \pm 1.21$ | $18.06 \pm 1.22$ | 2.67% |
| BODY | ACC | $89.19 \pm 1.76$ | $82.19 \pm 1.76$ | $76.17 \pm 1.26$ | $72.17 \pm 0.97$ | $91.63 \pm 0.09$ | $89.7 \pm 3.61$ | $89.59 \pm 1.17$ | - |
| | MRR | $33.19 \pm 0.76$ | $34.99 \pm 0.76$ | $10.27 \pm 0.47$ | $6.87 \pm 1.42$ | $36.75 \pm 0.42$ | $26.7 \pm 3.50$ | $34.93 \pm 1.07$ | - |
| TITLE | ACC | $89.76 \pm 1.76$ | $90.39 \pm 0.59$ | $80.48 \pm 0.49$ | $82.98 \pm 0.65$ | $93.58 \pm 0.15$ | $92.6 \pm 3.52$ | $*93.96 \pm 0.02$ | 0.41% |
| | MRR | $33.52 \pm 0.76$ | $30.25 \pm 1.76$ | $22.27 \pm 0.47$ | $18.87 \pm 1.42$ | $40.25 \pm 1.15$ | $40.1 \pm 4.82$ | $42.72 \pm 1.01$ | 6.14% |
| UCI | ACC | $73.52 \pm 1.02$ | $74.98 \pm 2.22$ | $76.12 \pm 2.98$ | $75.12 \pm 1.76$ | $80.13 \pm 1.14$ | $76.4 \pm 1.12$ | $*80.88 \pm 0.08$ | 0.94% |
| | MRR | $9.38 \pm 0.09$ | $9.12 \pm 0.37$ | $7.86 \pm 1.02$ | $7.65 \pm 0.46$ | $10.39 \pm 1.40$ | $9.2 \pm 4.32$ | $*10.99 \pm 0.92$ | 5.77% |
| ETH | ACC | $78.05 \pm 1.76$ | $78.12 \pm 1.76$ | $72.12 \pm 2.76$ | $77.02 \pm 1.48$ | $77.01 \pm 2.58$ | $62.6 \pm 1.73$ | $*85.10 \pm 1.46$ | 8.93% |
| | MRR | $30.98 \pm 0.62$ | $31.12 \pm 0.60$ | $32.12 \pm 0.73$ | $27.12 \pm 0.85$ | $35.68 \pm 1.30$ | $33.0 \pm 2.62$ | $38.08 \pm 1.57$ | 6.73% |
| | | | | | WINGNN SETTING | | | | |
| DATASET | METRIC | GCN-L | GCN-G | EGC-O | EGC-H | WINGNN | DEGNN | TMETANET | IMPR. |
| ALPHA | ACC | $59.72 \pm 1.98$ | $57.12 \pm 1.78$ | $58.91 \pm 1.98$ | $53.12 \pm 0.79$ | $83.15 \pm 0.51$ | $81.48 \pm 2.87$ | $*89.92 \pm 1.84$ | 8.14% |
| | MRR | $5.12 \pm 0.79$ | $4.12 \pm 0.79$ | $6.12 \pm 0.79$ | $3.12 \pm 0.79$ | $34.59 \pm 3.36$ | $32.36 \pm 0.90$ | $*38.93 \pm 3.06$ | 12.55% |
| OTC | ACC | $51.43 \pm 1.22$ | $50.78 \pm 2.89$ | $50.12 \pm 1.57$ | $50.12 \pm 0.97$ | $85.70 \pm 1.25$ | $81.87 \pm 0.08$ | $*90.43 \pm 1.17$ | 5.52% |
| | MRR | $11.43 \pm 1.25$ | $10.78 \pm 1.34$ | $10.12 \pm 1.25$ | $12.12 \pm 0.98$ | $37.92 \pm 2.16$ | $29.85 \pm 0.59$ | $*39.98 \pm 2.16$ | 5.43% |
| BODY | ACC | $70.42 \pm 3.76$ | $69.78 \pm 1.32$ | $77.34 \pm 3.76$ | $74.12 \pm 2.40$ | $97.43 \pm 1.40$ | OOM | $*98.26 \pm 1.29$ | 0.85% |
| | MRR | $3.56 \pm 1.32$ | $4.76 \pm 2.01$ | $5.43 \pm 2.32$ | $5.74 \pm 2.32$ | $16.56 \pm 4.54$ | OOM | $28.93 \pm 2.06$ | 74.70% |
| TITLE | ACC | $70.34 \pm 0.56$ | $72.01 \pm 2.98$ | $77.47 \pm 1.27$ | $83.45 \pm 1.87$ | $98.38 \pm 0.09$ | OOM | $**99.63 \pm 0.07$ | 1.27% |
| | MRR | $2.56 \pm 0.99$ | $2.76 \pm 2.01$ | $4.23 \pm 2.32$ | $3.88 \pm 3.32$ | $30.57 \pm 5.09$ | OOM | $34.96 \pm 2.06$ | 14.36% |
| UCI | ACC | $50.25 \pm 1.24$ | $50.82 \pm 3.01$ | $55.43 \pm 1.98$ | $56.12 \pm 1.98$ | $85.05 \pm 2.69$ | $75.11 \pm 1.02$ | $*86.37 \pm 5.63$ | 1.55% |
| | MRR | $7.25 \pm 3.99$ | $6.82 \pm 3.01$ | $10.43 \pm 1.87$ | $8.12 \pm 1.98$ | $20.94 \pm 3.78$ | $25.31 \pm 1.02$ | $*25.31 \pm 1.02$ | 20.87% |
| ETH | ACC | $60.98 \pm 1.37$ | $61.78 \pm 2.97$ | $66.43 \pm 5.37$ | $67.12 \pm 4.09$ | $95.62 \pm 1.92$ | OOM | $97.83 \pm 1.53$ | 2.31% |
| | MRR | $9.98 \pm 3.32$ | $11.78 \pm 1.92$ | $16.43 \pm 3.91$ | $17.12 \pm 4.07$ | $66.97 \pm 2.12$ | OOM | $78.07 \pm 1.09$ | 16.57% |

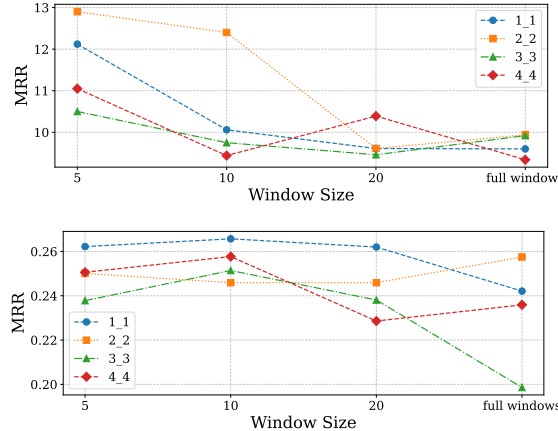

*Figure 3.* Impact of hyperparameters of zigzag persistence. The top figure shows the results under the ROLAND setting, while the bottom figure corresponds to the WinGNN setting.

that longer temporal aggregation dilutes informative signals and introduces noise. In the WinGNN setting, the performance is generally more stable and less sensitive to window size. The $(1, 1)$ configuration again achieves the highest peak MRR at $w = 10$, while the $(2, 2)$ configuration performs comparably well with smaller variance. Interestingly, even with large windows, most configurations maintain acceptable MRR values, indicating the WinGNN setting is more tolerant to extended temporal aggregation.

These findings highlight the importance of maintaining fine-grained temporal resolution and prioritizing local structural relationships when extracting topological features in dynamic graph scenarios.

### 6.4. Noise Robustness

We also conduct noise robustness experiments under different settings on the Reddit-Title dataset by using two types

Table 2. Ablation studies on the Alpha and UCI datasets.

| SETTINGS | METRIC | ALPHA | UCI |
|---|---|---|---|
| **LIVE UPDATE SETTING (ROLAND)** | | | |
| TOPO | ACC | *86.84 $\pm$ 1.02 | *80.88 $\pm$ 0.08 |
| | MRR | *17.68 $\pm$ 0.55 | *10.99 $\pm$ 0.92 |
| RANDOM | ACC | 83.07 $\pm$ 2.44 | 80.04 $\pm$ 1.26 |
| | MRR | 15.85 $\pm$ 0.63 | 10.32 $\pm$ 0.36 |
| DIST | ACC | 84.24 $\pm$ 1.42 | 79.96 $\pm$ 0.94 |
| | MRR | 15.55 $\pm$ 0.72 | 10.88 $\pm$ 0.87 |
| **WINGNN SETTING** | | | |
| TOPO | ACC | *89.92 $\pm$ 1.84 | *86.37 $\pm$ 5.63 |
| | MRR | *38.93 $\pm$ 3.06 | *25.31 $\pm$ 1.02 |
| RANDOM | ACC | 82.14 $\pm$ 1.76 | 83.83 $\pm$ 4.13 |
| | MRR | 33.98 $\pm$ 3.12 | 21.33 $\pm$ 4.92 |
| DIST | ACC | 88.63 $\pm$ 1.23 | 84.81 $\pm$ 2.31 |
| | MRR | 38.02 $\pm$ 2.92 | 24.20 $\pm$ 2.72 |

of noise injection methods, i.e., evasion attack, which contaminates only the test set, and poisoning attack, which contaminates both the training and test sets. Both methods involve selecting a certain proportion of nodes to flip edges (deleting existing edges and adding non-existing edges) with contamination ratios of 5%, 10%, 20%, and 30% respectively. From Figure 4, we can see that TMetaNet exhibits better stability than WinGNN under both poisoning and evasion attacks, demonstrating that Dowker Zigzag Persistence has good robustness when facing noise.

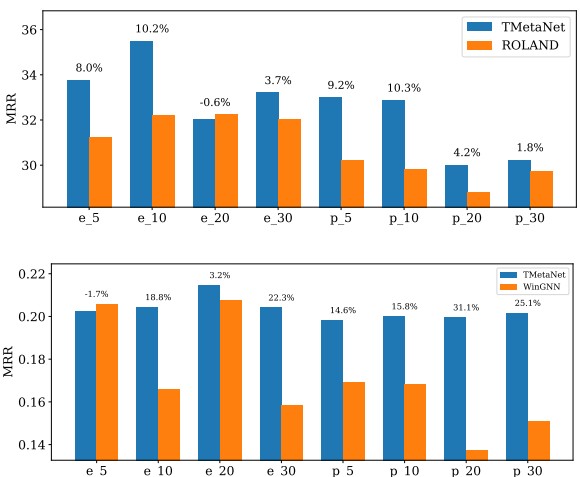

Figure 4. Robustness analysis of TMetaNet under different settings, where e_10 represents a 10% edge evasion attack. The top figure shows the results under the ROLAND setting, while the bottom figure corresponds to the WinGNN setting.

# 7. Conclusion

We have presented Dowker Zigzag Persistence (DZP), which systematically captures higher-order structural features in evolving graphs. To bridge topological analysis with deep learning, we have developed TMetaNet based on DZP, a novel topology-enhanced meta-learning framework for dynamic graph neural networks that integrates persistent homology with meta-learning. Comprehensive experiments on six real-world datasets have demonstrated TMetaNet's superior performance and robustness. Our techniques provide a new reliable and computationally efficient pathway for extracting higher-order topological features from dynamic graphs, bringing new insights into adaptive parameter updates for meta-learning.

## Acknowledgements

This work was supported by the National Key R&D Program of China under Grant Nos. 2024YFE0115900 and 2024YFB2908001. Chen and Gel have received no research support from any non-US-based entity. Chen and Gel are grateful for the travel support from the Institute of the Mathematical Sciences (IMS) of the National University of Singapore (NUS) to attend IMS-NTU joint workshop on Biomolecular Topology: Modelling and Data Analysis which has sparked some of the presented ideas.

## Impact Statement

We do not anticipate any negative impact from our topological meta-learning framework. Conversely, we posit that integrating persistent homology with adaptive meta-learning mechanisms establishes new foundations for reliable, explainable, and robust analysis of evolving network systems. Specifically, Dowker Zigzag Persistence enables principled extraction of multiscale topological signatures that capture essential structural dynamics while filtering transient noise - a critical capability for high-stakes applications like financial fraud detection and critical infrastructure monitoring. The meta-learning adaptation mechanism further enhances operational safety by dynamically adjusting model sensitivity to topological shifts, preventing overreaction to spurious patterns.

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

## A. Persistent Homology on Graphs

Specifically, given a graph $G = (V, E)$, the process of computing persistent homology can be described in the following two steps. **1) Construct the complex.** Based on a filtration function $f(G)$, we construct a sequence of nested subgraphs $G^0 \subseteq G^1 \subseteq \cdots \subseteq G^L$, and further construct the corresponding sequence of complexes $\mathcal{C}_0 \subseteq \mathcal{C}_1 \subseteq \cdots \subseteq \mathcal{C}_L = \mathcal{C}(G^L)$. **2) Compute the homology groups.** Based on the sequence of complexes, we compute their corresponding homology groups $\mathcal{H}_p(\mathcal{C}_i)$, $i, p \geq 0$, where $p$ denotes the dimension, and $\beta \in \mathcal{H}_p(\mathcal{C}_i)$ represents the $p$-dimensional topological features. As the sequence of complexes evolves, the homology groups reflect the evolution of topological features. Finally, it is represented in the form of a persistence diagram, $PD(\mathcal{C}) = \{(b_\beta, d_\beta) \mid \beta \in H_p(\mathcal{C})\}$, where $b_\beta, d_\beta$ represent the birth and death times of the topological features, respectively.

A common complex on graphs is the Vietoris-Rips Complex, defined as follows:

**Definition A.1** (Vietoris-Rips Complex on Graphs). Given a graph $G = (V, E)$ with graph distance $d_G$, the *Vietoris-Rips complex* at scale $\delta \geq 0$ is the simplicial complex defined by:

$$\mathrm{VR}_\delta(G) = \big\{ \sigma \subseteq V \mid \forall u, v \in \sigma, \ d_G(u, v) \leq \delta \big\} \tag{5}$$

where $d_G(u, v)$ denotes the shortest path distance between nodes $u$ and $v$. A $k$-simplex is included if and only if the diameter of its $(k+1)$ vertices (maximum pairwise distance) is at most $\delta$.

When faced with dynamic graphs, the Vietoris-Rips complex needs to be computed for all nodes at each time step, which leads to high computational complexity.

**Definition A.2** (Homological Dimension). The homological dimension $k$ of a simplicial complex $C$ is defined as:

$$k = \dim H_*(C) = \sup\{n \in \mathbb{N} \mid H_n(C) \neq 0\} \tag{6}$$

where $H_n(C)$ denotes the $n$-th homology group of $C$.

## B. Proof of Dowker Zigzag Persistence Stability

This section provides a detailed proof of the stability of **Dowker Zigzag Persistence (DZP)** in discrete-time dynamic graphs. The proof is based on the Theorem 3 in (Ye et al., 2023), showing that DZP remains stable under controlled perturbations of the underlying graph structure. Below we restate essential definitions and theorems, then show step-by-step arguments.

### B.1. Foundational Definitions

**Definition B.1** (Discrete-Time Dynamic Graph). A discrete-time dynamic graph $\mathcal{G} = \{G_0, G_1, ..., G_T\}$, where $G_t = (V_t, E_t)$, consists of:

- Time-indexed node sets $\{V_t\}_{t=1}^T$ where $V_t \subseteq \mathcal{V}$, $\mathcal{V}$ is the vertex set of $\mathcal{G}$

- Weight matrices $\{\omega_t : V_t \times V_t \to \mathbb{R}^+\}_{t=1}^T$

satisfying temporal consistency:

$$\forall t \in \{1, \ldots, T-1\}, \ V_t \cap V_{t+1} \neq \emptyset \tag{7}$$

This definition captures evolving graph structures with temporal connectivity through overlapping node sets across discrete time steps. In real-world networks, this is a common phenomenon.

**Definition B.2** (Tripod). A tripod $R$ between $\mathcal{G}^X$ and $\mathcal{G}^Y$ consists of:

- Intermediate set $W$ with surjections:

$$X \xleftarrow{\pi_1} W \xrightarrow{\pi_2} Y$$

- Temporal consistency: $\forall t \in \{1, \ldots, T\}$,

$$\pi_1^{-1}(V_t^X) = \pi_2^{-1}(V_t^Y) \tag{8}$$

Tripods establish temporal correspondence through shared intermediate nodes while maintaining alignment of node existence across time steps.

**Lemma B.3** (Tripod Composition). *Given tripods $\mathcal{G}^X \xrightarrow{R_1} \mathcal{G}^Y$ and $\mathcal{G}^Y \xrightarrow{R_2} \mathcal{G}^Z$, there exists a composite tripod $R = R_2 \circ R_1$ such that:*

$$\mathcal{G}^X \xrightarrow{R} \mathcal{G}^Z \tag{9}$$

*Proof.* Construct the fiber product:

$$W = \{(w_1, w_2) \in W_1 \times W_2 \mid \pi_2^{(1)}(w_1) = \pi_1^{(2)}(w_2)\}$$

with projections:

$$\pi_1^R(w_1, w_2) = \pi_1^{(1)}(w_1)$$
$$\pi_2^R(w_1, w_2) = \pi_2^{(2)}(w_2)$$

Temporal consistency follows from:

$$(\pi_1^R)^{-1}(V_t^X) = (\pi_2^R)^{-1}(V_t^Z)$$

$\square$

This lemma establishes the compositionality of tripods, which is crucial for building complex temporal graph alignments. The composite tripod $R$ is constructed through a fiber product that ensures the intermediate graph $\mathcal{G}^Y$ serves as a bridge between $\mathcal{G}^X$ and $\mathcal{G}^Z$. The temporal consistency is preserved through the alignment of nodes in the intermediate graph $\mathcal{G}^Y$, where $(\pi_2^{(1)})^{-1}(V_t^Y)$ and $(\pi_1^{(2)})^{-1}(V_t^Y)$ establish the correspondence between the temporal nodes of $\mathcal{G}^X$ and $\mathcal{G}^Z$. This composition operation allows us to chain multiple tripods together, enabling the alignment of temporal graphs across multiple domains while maintaining temporal consistency.

**Definition B.4** (Bottleneck Distance for Dowker Zigzag Persistence). Let $\mathrm{Dgm}_{DZP}(\mathcal{G}^X)$ and $\mathrm{Dgm}_{DZP}(\mathcal{G}^Y)$ denote the Dowker Zigzag persistence diagrams (or barcodes) arising from two dynamic graphs $\mathcal{G}^X = \{G_t^X\}_{t=1}^T$ and $\mathcal{G}^Y = \{G_t^Y\}_{t=1}^T$.

We define the bottleneck distance $d_B\big(\mathrm{Dgm}_{DZP}(\mathcal{G}^X), \mathrm{Dgm}_{DZP}(\mathcal{G}^Y)\big)$ as follows:

$$d_B\big(\mathrm{Dgm}_{DZP}(\mathcal{G}^X), \mathrm{Dgm}_{DZP}(\mathcal{G}^Y)\big) := \inf_{\mu} \sup_{x \in \mathrm{DZP}(\mathcal{G}^X)} \| x - \mu(x)\|, \tag{10}$$

where we view each zigzag persistence diagram as a multiset of points (or equivalently, interval endpoints) in a suitable metric space (often the extended plane), and $\mu$ ranges over all bijections (or partial matchings) between the points of $\mathrm{DZP}(\mathcal{G}^X)$ and $\mathrm{DZP}(\mathcal{G}^Y)$. The distance $\| \cdot \|$ typically denotes the $\ell^\infty$ metric on the plane when diagrams are depicted as (birth, death) points, but other equivalent definitions exist. In simpler terms, this quantity measures the smallest value $\rho$ such that one can pair off the points of the two diagrams (with unpaired points matched to the diagonal if needed) so that each pair lies within distance $\rho$.

### B.2. Discrete $\epsilon$-Smoothing

**Definition B.5** (Discrete $\epsilon$-Smoothing). For $\epsilon \in \mathbb{N}^+$ and $t \in \{1, \dots, T\}$, define:

$$S_\epsilon V_t := \bigcup_{\tau=\max(1,t-\epsilon)}^{\min(T,t+\epsilon)} V_\tau$$

$$S_\epsilon \omega_t[(x, x')] := \frac{1}{2\epsilon + 1} \sum_{\tau=t-\epsilon}^{t+\epsilon} \omega_\tau[(x, x')] \cdot \mathbf{1}_{\{x, x' \in V_\tau\}}$$

where $\mathbf{1}$ is the indicator function for temporal existence.

The smoothing operator $S_\epsilon$ aggregates node sets within temporal windows and averages edge weights with existence validation, ensuring meaningful comparisons across snapshots.

**Lemma B.6** (Smoothing Bijectivity). *For $\epsilon \leq \zeta$ (max smoothing radius), there exists an identity tripod $R_I$ satisfying:*

$$\mathcal{G}^X \xrightarrow{R_I} S_\epsilon \mathcal{G}^X$$

$$S_\epsilon \mathcal{G}^X \xrightarrow{R_I} \mathcal{G}^X$$

*Proof.* Take $W = X$ with identity maps $\pi_1 = \pi_2 = \mathrm{id}_X$. Temporal consistency holds because:

$$S_\epsilon V_X(t) \supseteq V_X(t) \quad \text{and} \quad \omega_X^t \leq S_\epsilon \omega_X^t$$

Bijectivity is preserved through the indicator function in Definition B.5. □

### B.3. Discrete $\epsilon$-Interleaving

**Definition B.7** (Discrete $\epsilon$-Interleaved Dynamic Graphs). Two discrete-time dynamic graphs $\mathcal{G}^X = \{V_t^X, \omega_t^X\}_{t=1}^T$ and $\mathcal{G}^Y = \{V_t^Y, \omega_t^Y\}_{t=1}^T$ are $\epsilon$-**interleaved** if there exist:

- Tripods $R_1, R_2$ satisfying Definition B.2
- Smoothing operator $S_\epsilon$ from Definition B.5

such that:

$$\mathcal{G}^X \xrightarrow{R_1} S_\epsilon \mathcal{G}^Y$$

$$\mathcal{G}^Y \xrightarrow{R_2} S_\epsilon \mathcal{G}^X$$

with discrete distortion measures:

$$\mathrm{dis}_V(R) = \max_{1 \leq t \leq T} \frac{|V_t^X \setminus \pi_1(R_t)| + |V_t^Y \setminus \pi_2(R_t)|}{|V_t^X| + |V_t^Y|}$$

$$\mathrm{dis}_\omega(R) = \max_{1 \leq t \leq T} \sup_{(x,y) \in R_t} |\omega_t^X(x) - \omega_t^Y(y)|$$

### B.4. Discrete Structural Metric

**Definition B.8** (Discrete Dynamic Graph Structural Metric). For two $\epsilon$-interleaved dynamic graphs $\mathcal{G}^X, \mathcal{G}^Y$, their structural distance is:

$$d_{\mathcal{G}}(\mathcal{G}^X, \mathcal{G}^Y) := \frac{1}{2} \inf_{R_1, R_2} \max \begin{cases} \mathrm{dis}_V(R_1) + \mathrm{dis}_V(R_2), \\ \mathrm{dis}_\omega(R_1) + \mathrm{dis}_\omega(R_2), \\ \max_t C_t(R_1, R_2) \end{cases} \tag{11}$$

where co-distortion measure:

$$C_t(R_1, R_2) = \max \begin{cases} \sup_{\substack{x \in V_t^X \\ y \in V_t^Y}} |S_\epsilon \omega_t^X(x, \psi_t(y)) - S_\epsilon \omega_t^Y(\phi_t(x), y)|, \\ \sup_{\substack{y \in V_t^Y \\ x \in V_t^X}} |S_\epsilon \omega_t^Y(y, \phi_t(x)) - S_\epsilon \omega_t^X(\psi_t(y), x)| \end{cases} \tag{12}$$

with $\phi_t = \pi_2 \circ \pi_1^{-1}$, $\psi_t = \pi_1 \circ \pi_2^{-1}$ being the induced mappings.

### B.5. $\varepsilon$-Net Interleaving Preservation

**Theorem B.9** ($\varepsilon$-Net Interleaving Inheritance). *Let $\mathcal{G}^X, \mathcal{G}^Y$ be $\epsilon$-interleaved dynamic graphs with $\varepsilon$-nets $L^X, L^Y$. If the $\varepsilon$-nets satisfy:*

$$\varepsilon \geq \epsilon + \delta_{\max} \tag{13}$$

*where $\delta_{\max} = \max_t \mathrm{diam}(S_\epsilon V_t)$, then $L^X$ and $L^Y$ are $\epsilon$-interleaved.*

*Proof.* **1. Natural Correspondence** By $\epsilon$-interleaving property, the original tripod $R$ induces mappings:

$$\forall t, \exists \phi_t : L_t^X \hookrightarrow S_\epsilon L_t^Y, \; \psi_t : L_t^Y \hookrightarrow S_\epsilon L_t^X \tag{14}$$

**2. Structural Preservation** For $l_x \in L_t^X$, using $\varepsilon$-net coverage:

$$\begin{aligned} d(l_x, \psi_t \circ \phi_t(l_x)) &\leq d(l_x, S_\epsilon L_t^X) + \epsilon \quad \text{(triangle inequality)} \\ &\leq \delta_{\max} + \epsilon \leq \varepsilon \quad \text{(by condition)} \end{aligned}$$

**3. Temporal Consistency** The $\epsilon$-smoothing operator ensures:

$$S_\epsilon L_t^X \subseteq \bigcup_{\tau=t-\epsilon}^{t+\epsilon} L_\tau^X \quad \text{and} \quad S_\epsilon L_t^Y \subseteq \bigcup_{\tau=t-\epsilon}^{t+\epsilon} L_\tau^Y \tag{15}$$

$\square$

**Theorem B.10** (Dowker Zigzag Persistence Stability). *For $\epsilon$-interleaved dynamic graphs $\mathcal{G}^X, \mathcal{G}^Y$ with structural metric $d_\mathcal{G}(\mathcal{G}^X, \mathcal{G}^Y)$, their Dowker Zigzag Persistence diagrams satisfy:*

$$d_B\left(\mathrm{Dgm}_{DZP}(\mathcal{G}^X), \mathrm{Dgm}_{DZP}(\mathcal{G}^Y)\right) \leq K \cdot d_\mathcal{G}(\mathcal{G}^X, \mathcal{G}^Y) \tag{16}$$

*where constant $K$ depends only on the homological dimension.*

*Proof.* By Theorem B.9, $\varepsilon$-nets inherit $\epsilon$-interleaving with:

$$d_\mathcal{L}(L^X, L^Y) \leq 2d_\mathcal{G} \tag{17}$$

where the factor 2 comes from:

$$\underbrace{1}_{\text{original}} + \underbrace{1}_{\text{smoothing compensation}} \tag{18}$$

**1. Construct Interleaving Diagrams**
Given $d_\mathcal{L}(L^X, L^Y) = \delta$, build commutative diagrams for each timestamp $t$:

$$\begin{array}{ccccc} D(L_t^X) & \longleftrightarrow & D(L_t^X \cup L_{t+1}^X) & \longleftrightarrow & D(L_{t+1}^X) \\ \Phi_t \downarrow & & \Psi_t \downarrow & & \Phi_{t+1} \downarrow \\ D(L_t^Y)^\delta & \longleftrightarrow & D(L_t^Y \cup L_{t+1}^Y)^\delta & \longleftrightarrow & D(L_{t+1}^Y)^\delta \end{array} \tag{19}$$

where vertical maps satisfy $\delta$-interleaving conditions.

**2. Verify $\delta$-Interleaving**
For any simplex $\sigma \in D(L_t^X)$, find matching $\sigma' \in D(L_t^Y)$ via:

$$\begin{aligned} \Phi_t(\sigma) &= \{\tau \in D(L_t^Y) \mid \exists \text{ witness } w : d(\sigma, w) \leq \delta\} \\ \Psi_t(\sigma') &= \{\tau \in D(L_t^X) \mid \exists \text{ witness } w' : d(\sigma', w') \leq \delta\} \end{aligned}$$

This guarantees:

$$\begin{cases} D(L_t^X) \hookrightarrow D(L_t^Y)^\delta \\ D(L_t^Y) \hookrightarrow D(L_t^X)^\delta \end{cases} \tag{20}$$

**3. Bottleneck Distance Bound.** In short, the classical statement in persistent homology is that if two (single-parameter) persistence modules are $\delta$-interleaved, then their bottleneck distance is at most $\delta$ (Cohen-Steiner et al., 2005). We provide a concise sketch of that reasoning below, then indicate how it generalizes to the zigzag case.

**Case: Single-parameter Filtration (classical statement).**    Consider two single-parameter filtrations $F_\alpha$ and $G_\alpha$ (for real parameter $\alpha$). They are called "$\delta$-interleaved" if for all $\alpha$:

$$F_\alpha \hookrightarrow G_{\alpha+\delta} \quad \text{and} \quad G_\alpha \hookrightarrow F_{\alpha+\delta}. \tag{21}$$

From these inclusions, one deduces: 1. Whenever a homology class is "born" in $F_\alpha$, it must appear in $G_{\alpha+\delta}$ at the latest (so its birth can shift by at most $\delta$). 2. Whenever that class "dies" in $F_\beta$, it disappears in $G_{\beta+\delta}$ at the latest (so its death can shift by at most $\delta$). Thus, a homology class that persists during the interval $[\alpha, \beta]$ in $F_\bullet$ will persist during an interval at most $[\alpha + \delta, \beta + \delta]$ in $G_\bullet$. Equivalently, the barcodes (interval decompositions) of $F_\bullet$ and $G_\bullet$ can be matched so that all intervals differ by at most $\delta$ in endpoints—this precisely implies their bottleneck distance is at most $\delta$.

**Case: Zigzag Filtration.**    For zigzag modules, we do not have a simple monotone filtration, but rather a sequence

$$F_1 \longleftrightarrow F_2 \longleftrightarrow \cdots \longleftrightarrow F_T \tag{22}$$

with forward/backward inclusions. While conceptually more involved, the same principle holds: if we have a $\delta$-interleaving between two zigzag modules (compatible inclusions that shift by at most $\delta$), then the birth and death of any homology class can shift by at most $\delta$. Hence the bottleneck distance of the two zigzag persistence modules is at most $\delta$.

**Dimension-dependent constant $K'$.**    Sometimes, one sees a constant factor $K'$ (depending on the dimension or the grading) in front of $\delta$. Intuitively, in higher-dimensional homology, identifying or mapping homology classes from one module to the other might require an extra factor. However, in many discrete or standard cases, we absorb these small local expansions into a constant $K'$ independent of the filtration size or the number of time steps. Thus we write

$$d_B(F_\bullet, G_\bullet) \leq K' \delta \leq K \cdot d_{\mathcal{G}}(\mathcal{G}^X, \mathcal{G}^Y), \tag{23}$$

where $K$ does not depend on $T$.                                                                                                                    □

# C. Method Details

## C.1. $\varepsilon$-net

In this section, we provide a detailed explanation of the construction method for the $\varepsilon$-net of a single snapshot, followed by the method to construct the $\varepsilon$-net for the entire dynamic graph based on the single snapshot $\varepsilon$-net construction method.

Taking the UCI dataset as an example, where $\varepsilon$ and $\delta$ represent the parameters for constructing landmarks, the statistics of landmarks under different parameter configurations are as follows:

|  | 1_1 | 2_2 | 3_3 | 4_4 |
|---|---|---|---|---|
| average proportion | 43% | 21% | 16% | 12% |
| average overlap rate | 35% | 22% | 17% | 16% |

## C.2. Zigzag Persistence Image

The zigzag persistence diagram space was equipped with distances derived from the Hausdorff metric. However, the structure of the metric space alone is insufficient as input for machine-learning techniques that require the inner product structure. To address this limitation, we transformed the ZPDs to zigzag persistence image (ZPI) representation, pixel arrays in Euclidean space. For each persistence diagram $Dgm_{n,k}(\mathcal{G}_t)$, we computed its ZPI .

$$\rho_{Dgm_{n,k}(\mathcal{G}_t)} = \sum_{\mu \in Dgm'_{n,k}(\mathcal{G}_t)} g(\mu) \exp\left(-\frac{\|z - \mu\|^2}{2\theta^2}\right), \tag{24}$$

$$Dgm'_{n,k}(\mathcal{G}_t)(x', y') = (x, y - x), \tag{25}$$

---

**Algorithm 1** Construct $\varepsilon$-net for Snapshots

---

**Input:** $dist\_matrix$ (The distance matrix representing the shortest path distances between nodes), $\varepsilon$ (A threshold value defining the minimum allowed distance between nodes in the $\varepsilon$-net), $nodes$ (A set of nodes to be considered for the $\varepsilon$-net), $mapped\_landmark\_seed$ (A set of predefined seed nodes to be prioritized for inclusion)

**Output:** $\varepsilon\_net$ (The constructed $\varepsilon$-net containing nodes whose pairwise distances are greater than $\varepsilon$)

  1: $covered\_nodes \leftarrow \emptyset$ {Set to keep track of selected nodes}
  2: $node\_degrees \leftarrow \emptyset$ {Dictionary to store the degree of each node}
  3: **for** each $node \in nodes$ **do**
  4:    $degree \leftarrow 0$
  5:    **for** each $i \in \{1, \ldots, |nodes|\}$ **do**
  6:      **if** $dist\_matrix[i, node] \leq \varepsilon \wedge i \neq node$ **then**
  7:        $degree \leftarrow degree + 1$
  8:      **end if**
  9:    **end for**
10:    $node\_degrees[node] \leftarrow degree$
11: **end for**
12: $sorted\_nodes \leftarrow$ Sort($nodes$ by $node\_degrees$ in descending order)
13: **if** $mapped\_landmark\_seed$ is not empty **then**
14:    $sorted\_seed \leftarrow$ Sort($mapped\_landmark\_seed$ by $node\_degrees$ in descending order)
15:    **for** each $start\_node \in sorted\_seed$ **do**
16:      $\varepsilon\_net \leftarrow \varepsilon\_net \cup \{start\_node\}$
17:      $covered\_nodes \leftarrow covered\_nodes \cup \{start\_node\}$
18:      **break** {Only include the highest-degree seed node first}
19:    **end for**
20:    **for** each $node \in sorted\_seed$ **do**
21:      **if** $node \notin covered\_nodes$ **then**
22:        **if** $\forall selected \in \varepsilon\_net, dist\_matrix[node, selected] > \varepsilon$ **then**
23:          $\varepsilon\_net \leftarrow \varepsilon\_net \cup \{node\}$
24:          $covered\_nodes \leftarrow covered\_nodes \cup \{node\}$
25:        **end if**
26:      **end if**
27:    **end for**
28: **end if**
29: **for** each $node \in sorted\_nodes$ **do**
30:    **if** $node \notin covered\_nodes$ **then**
31:      **if** $\forall selected \in \varepsilon\_net, dist\_matrix[node, selected] > \varepsilon$ **then**
32:        $\varepsilon\_net \leftarrow \varepsilon\_net \cup \{node\}$
33:        $covered\_nodes \leftarrow covered\_nodes \cup \{node\}$
34:      **end if**
35:    **end if**
36: **end for**
37: **return** $\varepsilon\_net$

---

---

**Algorithm 2** Construct $\varepsilon$-nets for Discrete-Time Dynamic Graphs

---

**Input:** $\mathcal{G} = \{G_t\}_{t=1}^{T}$ (The discrete-time dynamic graph), $\varepsilon$ (Threshold for the $\varepsilon$-net construction), $T$ (Number of time snapshots in the dynamic graph)

**Output:** Landmarks and Witnesses for all time snapshots

1: $landmarks_t \leftarrow \emptyset$ {Set to store landmarks for each snapshot}
2: $witnesses_t \leftarrow \emptyset$ {Set to store witnesses for each snapshot}
3: $mapped\_landmark\_seed \leftarrow \emptyset$ {Set to store the mapped landmarks from previous snapshot}
4: **for** $t = 1$ to $T$ **do**
5:    $nodes_t \leftarrow$ Get nodes from $G_t$ {Get the set of nodes for the current snapshot}
6:    $node\_degrees_t \leftarrow \emptyset$ {Dictionary to store degrees of nodes in current snapshot}
7:    **for** each $node \in nodes_t$ **do**
8:      $degree_t \leftarrow 0$
9:      **for** each $i \in \{1, \ldots, |nodes_t|\}$ **do**
10:        **if** $dist\_matrix_t[i, node] \leq \varepsilon \wedge i \neq node$ **then**
11:          $degree_t \leftarrow degree_t + 1$
12:        **end if**
13:      **end for**
14:      $node\_degrees_t[node] \leftarrow degree_t$
15:    **end for**
16:    $sorted\_nodes_t \leftarrow$ Sort($nodes_t$ by $node\_degrees_t$ in descending order)
17:    **if** $t = 1$ **then**
18:      $landmarks_t \leftarrow$ First few nodes from $sorted\_nodes_t$
19:      $witnesses_t \leftarrow$ Construct witnesses from $landmarks_t$ in $G_1$
20:    **else**
21:      $mapped\_landmark\_seed \leftarrow$ Landmarks from previous snapshot $G_{t-1}$
22:      $landmarks_t \leftarrow$ First few nodes from $sorted\_nodes_t$
23:      $witnesses_t \leftarrow$ Construct witnesses using $mapped\_landmark\_seed$ in $G_t$
24:    **end if**
25:    **Output** $landmarks_t$ and $witnesses_t$ for time snapshot $t$
26: **end for**
27: **return** All $landmarks_t$ and $witnesses_t$ for all time snapshots

---

where $Dgm'_{n,k}(\mathcal{G}_t)$ is the transformed multiset of $Dgm_{n,k}(\mathcal{G}_t)$. The weighting function $g(\mu)$ is defined with mean $\mu = (\mu_x, \mu_y) \in \mathbb{R}^2$ and variance $\theta^2$, which depends on the distance from the diagonal. $\rho_{Dgm_{n,k}(\mathcal{G}_t)}$ represents the zigzag persistence surface.

Next, we discretized a subdomain of the zigzag persistence surface $\rho_{Dgm_{n,k}(\mathcal{G}_t)}$ onto a grid. The ZPI, represented as a matrix of pixel values, is obtained by integrating over each grid cell. Specifically, the value of each pixel $z \in \mathbb{R}^2$ within the ZPI is defined as:

$$Z_{n,k}(z) = \iint_z \sum_{\mu \in Dgm'_{n,k}(\mathcal{G}_t)} g(\mu) exp\left\{ -\frac{\|x-\mu\|^2}{2\theta^2} \right\} dz_x dz_y, \tag{26}$$

$Z_{n,k}$ represents the $k$-dimensional ZPI under parameters $\varepsilon_n$ and $\delta_n$. This results in a matrix where each pixel value encapsulates the density of topological features within its corresponding grid cell, weighted by their proximity to the mean $\mu$.

### C.3. Toy Example of different task splitting strategies

Figure 5 illustrates a comparative analysis of distinct task splitting methodologies. ROLAND implements a snapshot-wise partitioning strategy, wherein each temporal snapshot $G_t$ is partitioned into training, validation, and testing subsets. The model undergoes training on the training subset of $G_{t-1}$, validation on the validation subset of $G_t$, and evaluation on the testing subset of $G_t$, thereby leveraging the complete temporal sequence for both training and testing phases. In contrast, WinGNN employs a chronological partitioning approach, segregating the temporal sequence into distinct training and testing periods. For instance, given a sequence comprising 6 temporal snapshots, WinGNN utilizes the initial 4 snapshots for training purposes and the remaining 2 snapshots for testing.

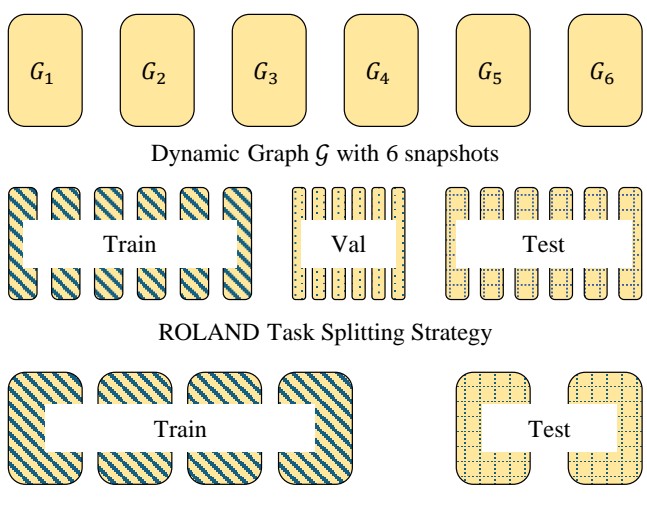

*Figure 5.* Toy Example of different task splitting strategies.

## D. Experimental Details

### D.1. Datasets Details

The datasets used in the experiments are summarized in Table 3.

### D.2. Baseline Details

The baseline models used in the experiments are:

| Dataset | # Edges | # Nodes | Range | # Snapshots |
|---|---|---|---|---|
| ETH-Yocoin (Li et al., 2020) | 746, 397 | 15, 682 | Jul 21 2016 - Feb 05 2018 | 426 |
| Reddit-Title (Kumar et al., 2018a) | 571,927 | 54,075 | Dec 31, 2013 - Apr 30, 2017 | 178 |
| Reddit-Body (Kumar et al., 2018a) | 286,561 | 35,776 | Dec 31, 2013 - Apr 30, 2017 | 178 |
| UCI-Message (Panzarasa et al., 2009) | 59,835 | 1,899 | Apr 15, 2004 - Oct 26, 2004 | 29 |
| Bitcoin-OTC (Kumar et al., 2018b) | 35,592 | 5,881 | Nov 8, 2010 - Jan 24, 2016 | 279 |
| Bitcoin-Alpha (Kumar et al., 2016) | 24,186 | 3,783 | Nov 7, 2010 - Jan 21, 2016 | 274 |

*Table 3.* Summary of the datasets used in our experiments.

- GCN-L and GCN-G: These combine structural encoding GCN modules with temporal encoding LSTM and GRU modules, widely used in dynamic link prediction. (Manessi et al., 2020)

- EGC-O and EGC-H: Use RNNs to update internal GNN parameters between snapshots, with EGC-O using LSTM encoding and EGC-H using GRU encoding. (Pareja et al., 2020)

- ROLAND: A meta-learning-based dynamic graph neural network that uses a live update training strategy, stacking a GCN module for capturing structural information and a GRU module for temporal encoding. (You et al., 2022)

- WinGNN: A meta-learning-based dynamic graph neural network that replaces explicit temporal encoding with in-window stochastic gradient aggregation. (Zhu et al., 2023b)

- DeGNN: A decoupled graph neural network that uses a two-stream architecture to capture both structural and temporal information. (Zheng et al., 2023)

### D.3. Experimental Settings

All experiments in this paper were conducted on a Linux server with 4 NVIDIA A6000 GPUs. For the ROLAND setting, we used the recommended parameter settings from ROLAND for the baselines. For the WinGNN setting, we used the recommended parameter settings from WinGNN for the baselines, except for GCN-L and GCN-G which were not used in WinGNN, so we selected their optimal parameter combinations through grid search. For TMetaNet, the core parameters include the following parts:

- Parameters for computing the Dowker Persistence Diagram. We choose $\varepsilon = 1$ and $\delta = 1$, with the window size set to full.

- Parameters for computing the Zigzag Persistence Image. We set the image size to be 50.

- Parameters for the TMetaNet meta-learning model, mainly including the meta-learning parameter update model's learning rate meta_lr and dropout rate. We use the grid search to select the optimal parameter combinations for each dataset.

*Table 4.* Running time comparison between TMetaNet and other meta-learning methods under different settings.

| LIVE UPDATE SETTING(ROLAND) | | | | | | |
|---|---|---|---|---|---|---|
| **Method** | **Alpha** | **OTC** | **Body** | **Title** | **UCI** | **ETH** |
| ROLAND | 0.49 | 0.53 | 1.21 | 1.90 | 0.75 | 0.43 |
| TMETANET | 1.04 | 1.15 | 2.58 | 4.15 | 1.81 | 1.45 |
| TIME INCREASE | 112% | 117% | 113% | 118% | 141% | 237% |
| **WINGNN SETTING** | | | | | | |
| WINGNN | 16.36 | 15.37 | 15.52 | 74.54 | 1.93 | 23.92 |
| TMETANET | 18.78 | 16.26 | 16.46 | 75.97 | 2.19 | 25.72 |
| TIME INCREASE | 14% | 6% | 6% | 2% | 13% | 7% |

## D.4. Running Time

We compare the running time of TMetaNet with other meta-learning methods under different settings. Please refer to Table 4. For ROLAND, the time shown is the average training time per snapshot, while for WinGNN, the time shown is the average training time per epoch. As can be seen, after adding the persistent homology-based meta-learning parameter adjustment, TMetaNet's running time increases. In the ROLAND setting, the model needs to run for multiple epochs at each snapshot, so we take the average training time per snapshot. Under the ROLAND setting, the meta-model parameter update needs to consider the parameter results from each epoch, resulting in a larger proportion of time increase. In the WinGNN setting, the model trains on the entire training set in each epoch, so we take the average training time per epoch. For meta-learning parameter updates within each epoch's snapshots, only one update is needed, resulting in a smaller proportion of time increase.

## D.5. Supplementary Experimental Results

*Table 5.* Supplementary dynamic link prediction performance of different models under different settings. The best results are highlighted in bold, and the second-best results are underlined. Each experiment is repeated five times, and the mean (omitting %) and standard deviation are reported. ∗ indicates statistical significance ($p - value < 0.05$ between TMetaNet and the second-best result).

### LIVE UPDATE SETTING(ROLAND)

| Dataset | Metric | GCN-L | GCN-G | EGC-O | EGC-H | ROLAND | TMetaNet | Impr. |
|---|---|---|---|---|---|---|---|---|
| ALPHA | AUC | 87.01 ± 1.00 | 89.72 ± 0.82 | 83.42 ± 0.78 | 85.64 ± 1.01 | 93.27 ± 1.00 | **93.93 ± 0.75** | 0.71% |
| | R@1 | 7.23 ± 0.50 | 7.01 ± 0.23 | 6.34 ± 0.89 | 6.12 ± 1.01 | 7.76 ± 0.80 | ***8.16 ± 0.49** | 5.15% |
| | R@3 | 14.99 ± 0.23 | 15.23 ± 0.47 | 13.27 ± 0.65 | 12.45 ± 0.45 | 16.60 ± 0.85 | ***17.17 ± 0.90** | 3.43% |
| | R@10 | 28.73 ± 1.23 | 29.87 ± 0.79 | 20.98 ± 0.99 | 25.82 ± 0.45 | 32.76 ± 0.84 | ***35.96 ± 0.93** | 9.77% |
| OTC | AUC | 80.12 ± 1.20 | 74.56 ± 1.78 | 73.45 ± 1.34 | 70.12 ± 1.02 | 93.36 ± 0.48 | ***93.52 ± 0.49** | 0.17% |
| | R@1 | 6.54 ± 0.96 | 6.01 ± 0.89 | 5.54 ± 0.78 | 4.53 ± 0.89 | 8.77 ± 0.76 | ***9.01 ± 1.05** | 2.74% |
| | R@3 | 13.54 ± 1.98 | 12.01 ± 1.82 | 9.54 ± 1.78 | 8.53 ± 1.89 | 18.45 ± 1.98 | **19.68 ± 1.82** | 6.67% |
| | R@10 | 27.54 ± 1.98 | 26.01 ± 1.82 | 23.54 ± 1.78 | 22.53 ± 1.89 | 36.47 ± 1.19 | **37.78 ± 1.74** | 3.59% |
| BODY | AUC | 88.77 ± 0.42 | 89.65 ± 0.35 | 82.77 ± 0.69 | 80.61 ± 0.42 | **96.81 ± 0.42** | 96.18 ± 0.25 | - |
| | R@1 | 18.00 ± 0.23 | 15.28 ± 0.15 | 12.90 ± 0.78 | 10.82 ± 0.67 | 24.10 ± 0.48 | ***27.35 ± 0.59** | 13.49% |
| | R@3 | 33.92 ± 1.20 | 30.13 ± 0.82 | 25.92 ± 0.76 | 20.98 ± 0.78 | 42.42 ± 0.42 | **42.77 ± 0.58** | 0.83% |
| | R@10 | 52.23 ± 0.20 | 49.23 ± 1.00 | 39.20 ± 0.89 | 33.90 ± 0.23 | **62.28 ± 0.37** | 62.17 ± 0.28 | - |
| TITLE | AUC | 94.71 ± 0.03 | 95.02 ± 0.03 | 92.30 ± 0.03 | 90.35 ± 0.02 | **97.93 ± 0.00** | 97.64 ± 0.00 | - |
| | R@1 | 22.34 ± 0.23 | 20.98 ± 0.72 | 14.57 ± 0.76 | 12.28 ± 0.46 | 26.34 ± 1.30 | ***30.90 ± 1.02** | 17.31% |
| | R@3 | 40.28 ± 2.03 | 41.23 ± 1.28 | 37.28 ± 1.72 | 35.32 ± 1.24 | 47.23 ± 1.19 | **48.33 ± 0.95** | 2.33% |
| | R@10 | 56.88 ± 0.87 | 57.88 ± 0.66 | 52.32 ± 2.98 | 49.23 ± 1.89 | **68.32 ± 0.71** | 68.30 ± 0.35 | - |
| UCI | AUC | 86.27 ± 0.74 | 83.23 ± 0.67 | 81.29 ± 0.93 | 80.81 ± 0.61 | 88.96 ± 0.51 | **88.98 ± 0.60** | 0.02% |
| | R@1 | 3.45 ± 0.93 | 3.22 ± 0.73 | 2.99 ± 1.09 | 2.38 ± 0.78 | 4.63 ± 1.13 | ***5.30 ± 0.67** | 14.47% |
| | R@3 | 8.00 ± 0.92 | 7.77 ± 0.81 | 5.16 ± 0.87 | 4.72 ± 0.76 | 9.49 ± 1.80 | ***10.28 ± 1.14** | 8.32% |
| | R@10 | 20.82 ± 1.01 | 15.27 ±1.87 | 12.23 ± 1.05 | 11.28 ± 0.98 | 21.51 ± 2.35 | ***22.34 ± 2.05** | 3.86% |
| ETH | AUC | 90.23 ± 1.10 | 88.92 ± 0.72 | 87.92 ± 1.98 | 85.89 ± 1.34 | 94.41 ± 1.06 | **94.75 ± 1.13** | 0.36% |
| | R@1 | 18.72 ± 0.93 | 19.92 ± 1.92 | 17.82 ± 2.03 | 15.37 ± 1.28 | 28.94 ± 1.62 | ***32.58 ± 1.61** | 12.58% |
| | R@3 | 35.29 ± 3.94 | 30.29 ± 1.82 | 24.23 ± 1.96 | 20.92 ± 0.82 | 38.49 ± 1.57 | **40.17 ± 1.67** | 4.36% |
| | R@10 | 38.28 ±2.23 | 40.92 ± 0.23 | 34.82 ± 1.37 | 34.29 ± 0.92 | **48.51 ± 0.79** | 48.42 ± 1.39 | |

### WinGNN setting

| DATASET | METRIC | GCN-L | GCN-G | EGC-O | EGC-H | WinGNN | TMetaNet | Impr. |
|---|---|---|---|---|---|---|---|---|
| ALPHA | AUC | 72.32 ± 0.65 | 70.82 ± 1.24 | 62.78 ± 1.18 | 69.83 ± 1.57 | 81.57 ± 1.18 | ***94.50 ± 1.45** | 15.85% |
| | R@1 | 1.32 ± 0.05 | 1.82 ± 0.04 | 1.23 ± 0.06 | 1.76 ± 0.37 | **23.26 ± 1.56** | 18.73 ± 1.49 | - |
| | R@3 | 4.32 ± 0.15 | 4.82 ± 0.12 | 3.97 ± 0.57 | 2.76 ± 0.47 | **43.21 ± 1.59** | 39.40 ± 1.61 | - |
| | R@10 | 8.23 ± 0.32 | 7.82 ± 0.49 | 7.12 ± 2.72 | 6.12 ± 1.07 | 65.82 ± 1.64 | ***75.31 ± 1.90** | 14.42% |
| OTC | AUC | 52.83 ± 1.34 | 56.23 ± 1.23 | 59.83 ± 1.57 | 52.78 ± 1.18 | 87.38 ± 1.21 | ***89.14 ± 2.22** | 2.01% |
| | R@1 | 3.16 ± 0.17 | 3.45 ± 0.12 | 5.01 ± 0.71 | 6.23 ± 0.48 | 20.76 ± 1.47 | ***22.51 ± 3.93** | 8.43% |
| | R@3 | 9.16 ± 0.12 | 10.45 ± 0.12 | 13.01 ± 1.01 | 14.01 ± 0.77 | **49.96 ± 1.05** | 49.15 ± 4.90 | - |
| | R@10 | 15.82 ± 4.92 | 16.90 ± 3.45 | 19.67 ± 3.47 | 20.67 ± 3.47 | 73.95 ± 1.10 | ***80.33 ± 2.21** | 8.63% |
| BODY | AUC | 72.01 ± 3.20 | 76.34 ± 2.85 | 81.34 ± 3.45 | 80.24 ± 2.67 | 99.34 ± 0.07 | **99.54 ± 0.27** | 0.20% |
| | R@1 | 1.01 ± 0.01 | 0.99 ± 0.02 | 1.99 ± 0.12 | 2.24 ± 2.67 | 5.39 ± 3.38 | ***11.57 ± 4.26** | 114.66% |
| | R@3 | 2.01 ± 0.12 | 2.12 ± 0.12 | 3.99 ± 0.12 | 4.24 ± 2.67 | 15.81 ± 2.28 | **24.05 ± 2.68** | 52.12% |
| | R@10 | 5.12 ± 0.11 | 6.12 ± 0.45 | 7.99 ± 0.12 | 8.24 ± 2.67 | 43.78 ± 3.01 | **46.29 ± 2.52** | 5.73% |
| TITLE | AUC | 88.45 ± 4.61 | 87.01 ± 3.48 | 97.03 ± 0.01 | 94.29 ± 0.12 | 99.49 ± 1.22 | **99.90 ± 0.01** | 0.41% |
| | R@1 | 0.00 ± 0.00 | 0.01 ± 0.00 | 0.01±0.00 | 0.01±0.00 | 16.65 ± 1.45 | **21.87 ± 0.47** | 31.35% |
| | R@3 | 0.54 ± 0.00 | 0.87 ± 0.03 | 1.01±0.00 | 0.54±0.00 | 34.56 ± 3.88 | **38.48 ± 0.89** | 11.34% |
| | R@10 | 1.22 ± 0.05 | 1.23 ± 0.02 | 2.82 ± 2.45 | 5.08 ± 1.68 | 60.62 ± 1.10 | **62.94 ± 2.92** | 3.83% |
| UCI | AUC | 56.72 ± 2.34 | 54.01 ± 1.32 | 65.08 ± 3.91 | 71.54 ± 2.89 | **94.37 ± 0.21** | 91.19 ± 1.83 | - |
| | R@1 | 3.45 ± 0.19 | 2.03 ± 0.02 | 7.01 ± 0.18 | 4.41 ± 0.06 | 12.56 ± 2.94 | **14.95 ± 1.31** | 19.03% |
| | R@3 | 8.67 ± 0.12 | 11.02 ± 0.23 | 10.54 ± 0.43 | 6.90 ± 0.48 | 22.20 ± 1.41 | **27.72 ± 1.56** | 24.86% |
| | R@10 | 15.76 ± 1.87 | 18.81 ± 0.76 | 15.92 ± 0.99 | 15.08 ± 1.99 | 40.62 ± 1.51 | ***45.13 ± 1.46** | 11.10 |
| ETH | AUC | 89.76 ± 2.45 | 87.98 ± 1.82 | 82.98 ± 2.34 | 85.67 ± 1.01 | 95.08 ± 0.01 | ***99.60 ± 0.01** | 4.75% |
| | R@1 | 13.56 ± 2.09 | 12.34 ± 2.19 | 15.92 ± 0.89 | 14.02 ± 1.23 | 54.87 ± 1.21 | ***63.81 ± 1.87** | 16.30% |
| | R@3 | 24.34 ± 2.97 | 25.98 ± 3.58 | 28.97± 2.23 | 25.78 ± 3.23 | 67.08 ± 1.83 | ***92.04 ± 1.27** | 37.21% |
| | R@10 | 42.38 ± 1.84 | 42.77 ± 1.70 | 45.98 ± 2.34 | 40.98 ± 3.45 | 80.08 ± 0.78 | ***98.97 ± 0.45** | 23.59% |

