# OpenReview forum: "TMetaNet: Topological Meta-Learning Framework for Dynamic Link Prediction"
_ICML.cc/2025/Conference — ICML 2025 poster_

### Official Review · Reviewer_rS6w · 2025-03-02

**Overall Recommendation:** 3

**Summary:**

The authors propose a meta-learning framework that leverages the topological information to guide parameter updates of GNN for dynamic graphs. Specifically, it uses the epsilon-net algorithm to select a set of landmark nodes from the complete graph and construct the Dowker Complex. Then it can use the DZP to capture the key topological properties of the dynamic graph and guide the parameter updates. Experimental results show the proposed method outperforms the SOTA baseline up to 74.70%.

**Claims And Evidence:**

The claims are found to be supported by clear and convincing evidence.

**Essential References Not Discussed:**

No essential references are found not discussed.

**Experimental Designs Or Analyses:**

The experiments are extensive and rigorous. The main experiments compare the proposed with 6 baseline methods, on 6 datasets, under two settings (Rol. and Win.), and in terms of 6 metrics. Standard deviation is presented and statistical significance is presented. A possible weakness is some of the baselines are not very new, and the results can be more convincing if more methods are in 2021-2024 (unless no such methods are remarkable).
Ablation, sensitivity, and robustness studies are presented clearly. The question is, why they are only conducted under the WinGNN setting but not the ROLAND?

**Methods And Evaluation Criteria:**

Yes, they make sense.
- Topology is an important aspect when analyzing graphs, and focusing on topology meta-learning should be effective and beneficial.
- Evaluation datasets cover both web community and financial domain and vary in size and snapshot amount.

**Other Comments Or Suggestions:**

1. As a submission to a conference related to computer science, the paper could give more space to address how the algorithm is implemented. For example, move the content in the Appendix to the main sections. This will make the paper more coherent and help readers understand how the method is implemented
2. The upper part of Figure 2 (G1, D1, G1∪G2) or similar illustration can be additionally presented near the background section to help understand the important concepts.

**Other Strengths And Weaknesses:**

Strengths:
s1: the time complexity is better than existing methods.
s2: topology is an important aspect of graph learning and should be considered in graph meta-learning
s3: experiments are extensive (see Experimental Designs Or Analyses)

Weaknesses:
w1: some of the baselines are not very new (see Experimental Designs Or Analyses)
w2: some writing is not very clear (see Other Comments Or Suggestions)

**Questions For Authors:**

What proportion of nodes are selected as landmarks and witnesses in some exemplary cases? Can the authors present some examples?

**Relation To Broader Scientific Literature:**

TMetaNet proposes to use the topology features of graphs for meta-learning and updating the parameters of dynamic GNN. It is a proper and significant extension based on existing literature.

**Theoretical Claims:**

Mathematical formulations in sections 3-5 are checked but not very closely. They seem to be conherent and could support the proposed method but their detailed correctness is not rigorously verified by the reviewer.

---

> ### Author Rebuttal · Authors · 2025-04-01
>
> **Q1**: A possible weakness is some of the baselines are not very new. Ablation, sensitivity, and robustness studies under the ROLAND setting ?
>
> **A1**:We have added DeGNN [1] (ICLR23') as a baseline, with experimental results shown below. ∗ indicates statistical significance (p-value < 0.05).
>
> *ROLAND setting*
> |||ALPHA|OTC|BODY|TITLE|UCI|ETH|
> |-|-|-|-|-|-|-|-|
> |DeGNN|ACC|76.7±1.33|76.9±1.33|89.7±3.6|92.6±3.5|76.4±1.12|62.6±1.73|
> ||MRR|12.5±1.03|15.0±1.21|26.7±3.5|40.1±4.8|9.2±4.3|33.0±2.62|
> |TMetaNet|ACC|86.84±1.02*|85.89±1.22*|89.59±1.17|93.96±0.02*|80.88±0.08*|85.10±1.46*|
> ||MRR|17.68±0.55*|18.06±1.22*|34.93±1.07*|42.72±1.01*|10.99±0.92*|38.08±1.57*|
>
> *WinGNN setting*
> |||ALPHA|OTC|BODY|TITLE|UCI|ETH|
> |-|-|-|-|-|-|-|-|
> |DeGNN|ACC|81.48±2.87|81.87±0.08|OOM|OOM|75.11±1.02|OOM|
> ||MRR|32.36±0.90|29.85±0.59|OOM|OOM|20.15±1.13|OOM|
> |TMetaNet|ACC|89.92±1.84*|90.43±1.17*|98.26±1.29*|99.63±0.07*|86.37±5.63*|97.83±1.53*|
> ||MRR|38.93±3.06*|39.98±2.16*|28.93±2.06*|34.96±2.06*|25.31±1.02*|78.07±1.09*|
>
> We are currently implementing [2] and will post the results when they are ready. The results demonstrate that TMetaNet outperforms DeGNN under both ROLAND and WinGNN settings, validating the TMetaNet effectiveness of TMetaNet for dynamic graph learning.
>
> [1] Decoupled Graph Neural Networks for Large Dynamic Graphs
> [2] SEGODE: a structure-enhanced graph neural ordinary differential equation network model for temporal link prediction
>
> Additionally, we have supplemented the Ablation, sensitivity, and robustness experiments under the Roland setting, with results shown below.
>
> *Ablation*
> |SETTINGS|METRIC|ALPHA|UCI|
> |-|-|-|-|
> |TOPO|ACC|86.84±1.02|80.88±0.08|
> ||MRR|17.68±0.55|10.99±0.92|
> |RANDOM|ACC|83.07±2.44|80.04±1.26|
> ||MRR|15.85±063|10.32±0.36|
> |DIST|ACC|84.24±1.42|79.96±0.94|
> ||MRR|15.55±0.72|10.88±0.87|
>
> *Sensitivity MRR*
> ||5|10|20|full|
> |-|-|-|-|-|
> |1_1|12.12|10.06|9.61|9.60|
> |2_2|12.90|12.40|9.61|9.94|
> |3_3|10.50|9.75|9.46|9.92|
> |4_4|11.05|9.44|10.39|9.34|
>
> *Robustness MRR*
> ||e_5|e_10|e_20|e_30|p_5|p_10|p_20|p_30|
> |-|-|-|-|-|-|-|-|-|
> ROLAND|33.76|34.49|32.03|33.20|32.99|32.87|30.01|30.23|
> TMetaNet|31.25|32.21|32.23|32.01|30.21|29.80|28.81|29.71|
>
> These findings are consistent with the results under the WinGNN setting, proving that TMetaNet is also effective under the Roland setting.
>
> **Q2**: As a submission to a conference related to computer science, the paper could give more space to address how the algorithm is implemented. For example, move the content in the Appendix to the main sections. This will make the paper more coherent and help readers understand how the method is implemented.
>
> **A2**:Thank you for your suggestion. We will adjust the paper layout to accommodate more algorithmic details.
>
> **Q3**: The upper part of Figure 2 (G1, D1, G1∪G2) or similar illustration can be additionally presented near the background section to help understand the important concepts.
>
> **A3**: In Fig 2 (in the main body), $D_1$ should actually be $D(G_1)$, representing the Dowker complex corresponding to graph $G_1$, and we have updated the figure accordingly. Additionally, we have added the definition of $D(G_t)$ in Definition 4.1. In this paper, $D(G_t)$ is equivalent to $D(L_t,W_t)$. The definition of $G_1 \cup G_2$ has been added to the background section, representing the intersection of two adjacent graphs.
>
> **Q4**:What proportion of nodes are selected as landmarks and witnesses in some exemplary cases? Can the authors present some examples?
>
> **A4**:Section 6.3 of the paper discusses DZP's parameter sensitivity on the UCI dataset, where ε and δ represent the parameters for constructing landmarks. The statistics of landmarks under different parameters are as follows:
>
> ||1_1|2_2|3_3|4_4|
> |-|-|-|-|-|
> |average proportion|43%|21%|16%|12%|
> |average overlap rate|35%|22%|17%|16%|
>
> As can be seen from Figure 3 in the main body, the experimental results corresponding to 1_1 are generally better, indicating that too few landmarks will reduce the ability to capture higher-order structures.

---

### Official Review · Reviewer_AgPX · 2025-03-08

**Overall Recommendation:** 2

**Summary:**

The paper proposes TMetaNet, a topological meta-learning framework for dynamic link prediction. Key contributions include:
(1) Dowker Zigzag Persistence (DZP): A method combining Dowker complexes and zigzag persistence to efficiently capture high-order topological features in dynamic graphs.
(2) TMetaNet Architecture: Integrates DZP into a meta-learning framework where a CNN-based adaptor adjusts learning rates using topological differences between graph snapshots.
The empirical validation results show that the proposed method outperforms baselines (e.g., ROLAND, WinGNN) on six datasets (e.g., Bitcoin-OTC, Reddit) in accuracy, MRR, and robustness to noise, with up to 74.7% improvement in MRR.

## update after rebuttal
I am not an expertise in computational topology, thus I am not quite familiar with the utilization of Lipschitz continuity assumption and various settings. Therefore I will just keep my score.

**Claims And Evidence:**

Some claims are supported by the experiments and ablation studies. Evidence includes:

1. Tables showing TMetaNet’s superior performance (e.g., 93.96% accuracy vs. 93.58% for ROLAND).

2. Ablation studies confirming the utility of topological features over random/noise-injected variants.

3. Noise robustness tests (evasion/poisoning attacks) showing stable performance.
Limitations: Theoretical proofs rely on external work (Ye et al., 2023); real-world noise validation is limited to synthetic perturbations.

Yet following important claim need to be clarified:
- What is high-order graph information / graph structures? How do they help in this work.

**Essential References Not Discussed:**

1. Neural ODEs (Chen et al., 2018) for continuous-time dynamics.
2. SNPE (Greenberg et al., 2019) in simulation-based inference.

**Ethical Review Concerns:**

N.A.

**Experimental Designs Or Analyses:**

Yes, I checked the soundness/validity. Experiments cover multiple settings (live update, WinGNN), noise scenarios, and hyperparameter sensitivity. Variance in results (e.g., ETH dataset MRR: ±1.57) is reported but significance is not deeply analyzed.

**Methods And Evaluation Criteria:**

Methods: DZP reduces computational complexity via landmark sampling (ε-nets); TMetaNet uses topological signatures to guide meta-learning updates.

Evaluation Criteria: Standard metrics (AUC, MRR) on benchmark datasets (Bitcoin, Reddit). Baselines include EvolveGCN, ROLAND, and WinGNN.

**Other Comments Or Suggestions:**

1. It would be a better practice to have the citations in the same bracket in chronological order.
2. Please check the equations, as some of them run over the margins.

**Other Strengths And Weaknesses:**

Strengths:
1. Novel integration of topology and meta-learning;
2. Rigorous benchmarking;
3. Noise robustness.

Weakness:
1. Theoretical reliance on external proofs;
2. Limited exploration of real-world noise

**Questions For Authors:**

1. In the abstract it is unclear that why we ought to take care of the intrinsic complex high-order topological information of evolving graphs. What is it connection with the dynamic link prediction.
2. What is high-order structural information? Why do we have to use it in this work? It remains unknown in the introduction and created a huge challenge of getting pace with the motivation of this work.
3. Since we target at using higher-order graph to help dynamic link prediction, any other method would work for finding higher-order graphs? Such ranging from simple correlation methods to complicated like relational inference / structural inference (e.g., Kipf et al. Neural Relational Inference for Interacting System, 2019; Löwe et al. Amortized Causal Discovery: Learning to Infer Causal Graphs from Time-Series Data, 2022; Zheng et al. Diffusion model for relational inference, 2024; Wang & Pang, Structural Inference with Dynamics Encoding and Partial Correlation Coefficients, 2024)?

**Relation To Broader Scientific Literature:**

This work builds on (1) meta-learning: extends ROLAND/WinGNN by incorporating topology, (2) persistent homology: as it uses zigzag persistence for dynamic graphs, improving scalability via Dowker complexes, and (3) dynamic GNNs: as it compares with EvolveGCN and GCN variants.

**Theoretical Claims:**

The stability proof (Theorem 4.5) cites external work but adapts it to DZP. The appendix provides a detailed proof sketch using ε-interleaving and bottleneck distance. Assumptions (e.g., Lipschitz continuity) are reasonable but not fully self-contained.

---

> ### Author Rebuttal · Authors · 2025-04-01
>
> **Q1**: High-order graph information.
>
> **A**: For higher-order information, we mean various types of graph (sub)structures formed by interactions of multiple nodes simultaneously. Why is this information important and when? Suppose, we design a certain fraudulent scheme for money laundering. To conceal these illegal activities, it is unlikely that only two nodes would be involved (it’ll be easier to identify fraud!). Indeed, to hide criminal traces, money laundering schemes include many parties (nodes). How to identify particular multi-node patterns (i.e., higher-order information) that may be important? We can leverage the tools of persistent homology (PH). PH looks at the graph at multiple resolutions and tracks when specific multi-node patterns (described by simplices of various orders) appear or disappear as we monotonically change resolution scales. Those topological patterns staying with us longer are likelier to be important. (Note the appearance of weird patterns may serve, e.g., as a signal of money laundering.) Generally, wherever the problem at hand can be characterized by the inherent importance of such intrinsic multi-node interactions, e.g., link prediction due to social or protein-protein interactions, extracting higher-order structural information and integrating it into ML model will likely help. Other tasks, such as graph classification may benefit less. What we are doing in this paper is that extract the most essential time-evolving higher-order information for link prediction in dynamic networks and also use it to guide parameter updates in meta-learning models, which  to the best of our knowledge has never been done before.
>
> **Q2**: The stability proof \& assumptions.
>
> **A**: We follow the standard practice in pure mathematics, where each statement in the derivation chain is either justified by previously published results or if it is newly derived, it is properly placed in a context of the previously obtained and published results. Specifically, we have explicitly restated the Lipschitz continuity assumption within our discrete-time DZP framework, and the ε-interleaving and bottleneck distance bound are also adapted to our discrete DZP setting. The theoretical results Theorem B.9, B.10 are new and have not been appeared before.
>
> **Q3**: Significance.
>
> **A**: TMetaNet learns learning rates based on corresponding models under different settings (ROLAND & WinGNN). Compared to ROLAND, TMetaNet is more suitable for WinGNN which learns from all snapshots within a certain window length which is also reflected in significance, we have more significant results under WinGNN.
>
> **Q4**: Essential References.
>
> **A**: Neural ODEs and Graph Neural ODEs describe dynamical processes on graphs from the perspective of dynamical systems, while SNPE approaches graph probability distributions from a probabilistic inference perspective on high-dimensional data. Our work starts from a known graph structure and leverages higher-order graph topology in a form of time-varying Dowker zigzag persistence, to enhance link prediction performance on dynamic graphs. Given the recent premise of simplicial models for dynamic systems on networks, we believe that combining persistent homology with methods like Graph Neural ODEs would be a highly promising future direction, we will add all suggested papers to Related Work section.
>
> **Q5**: Limited exploration of real-world noise.
>
> **A**: Our robustness experiments consider the widely used structural Evasion and Poisoning attacks, and we have added the new Reddit-title experiment under the ROLAND setting. The below table shows that TMetaNet still maintains competitive robustness under the ROLAND setting.
>
> Robustness MRR
> ||e_5|e_10|e_20|e_30|p_5|p_10|p_20|p_30|
> |-|-|-|-|-|-|-|-|-|
> ROLAND|33.76|34.49|32.03|33.20|32.99|32.87|30.01|30.23|
> TMetaNet|31.25|32.21|32.23|32.01|30.21|29.80|28.81|29.71|
>
> **Q6**: Citations and eqs.
>
> **A**: We have made the changes in the paper.
>
> **Q7**: Higher-order graphs \& dynamic link prediction.
>
> **A**: As noted above, we focus on dynamic higher-order graph information, induced by simultaneous multi-node interactions/interdependencies. Correlation methods focus inherently on linear dependencies and are not feasible for assessment of such joint multi-node interdependencies. The suggested papers relate to dynamic systems, assessment of Granger causality in time series using graphs, relational inference on time series through diffusion generative modeling, and structural inference. These papers neither consider higher-order multi-node graph information, nor dynamic link prediction. (However, we believe that our ZDP can potentially be integrated with structural inference). The closest approach to our ZDP is to use network motifs. However, in contrast to ZDP, network motifs are essentially ad-hoc, typically limited to 4-nodes only, are not easily generalizable for weighted networks, and do not enjoy important mathematical properties such as stability.

---

### Official Review · Reviewer_GZjY · 2025-03-13

**Overall Recommendation:** 3

**Summary:**

The paper introduces TMetaNet, a meta-learning framework leveraging topological information for dynamic link prediction. The authors integrate Dowker Zigzag Persistence with graph neural networks to capture evolving topological structures. The work demonstrates competitive performance across six datasets compared to state-of-the-art methods.

## update after rebuttal:
Thank the authors for their detailed responses. Most of my initial concerns have been addressed. As a result, I kept my original score.

**Claims And Evidence:**

No issue here.

**Essential References Not Discussed:**

N/A

**Experimental Designs Or Analyses:**

The experimental design is thorough, including baseline comparisons, ablation studies, and detailed analyses that effectively demonstrate the method's capabilities.

**Methods And Evaluation Criteria:**

The authors identify the challenge of accounting for structural evolution in parameter updates but address this primarily through topology-informed learning rates. This approach, while innovative, may not fully address the complexity of evolving graph structures.

The motivation for employing adaptive learning rates derived from topological features requires more explanation or stronger theoretical justification. The connection among learning rate adaptation, capturing structural evolution patterns, and downstream tasks could be more rigorously established.

**Other Comments Or Suggestions:**

The authors adopted two settings to train the model and the results are clearly provided. According to Table 1, TMetaNet's performance under "WinGNN Setting" is significantly better than that under "Live Update Setting" consistently while the other methods' performance are not. This phenomenon requires deeper analysis.

**Other Strengths And Weaknesses:**

The paper is well-structured with a clear exposition of background, methodology, theoretical foundations, and empirical validation.

**Questions For Authors:**

Are there any existing works that use PH for dynamic graphs?

**Relation To Broader Scientific Literature:**

The integration of topological analysis with deep learning through traditional persistent homology presents a novel approach to incorporating structural information into graph neural networks.

**Theoretical Claims:**

No

---

> ### Author Rebuttal · Authors · 2025-04-01
>
> **Q1**: This approach, while innovative, may not fully address the complexity of evolving graph structures.
>
> **A1**: We agree that our approach may not fully address the full complexity of evolving graph structures. However, we argue that given the currently existing methods, we achieve almost as much as possible. In particular, by using zigzag persistence, our proposed model inherently captures higher-order topological structures across different timestamps in both spatial and temporal dimensions. Additionally, Dowker complex can efficiently handle large-scale graphs by focusing on more compact and expressive components of the graph topology, resulting in higher scalability, comparing to more conventional persistence homology methods for graphs (and scalability nowadays is arguably one of the primary roadblocks on adopting PH tools, especially, for dynamic scenarios).  Considering the uncertainty and complexities of evolving graph structures, zigzag persistence and Dowker complex provide stable representations of the graph’s topology even in presence of noise or incomplete data. Certainly, our approach can be extended further, for example, by using zigzag persistence along multiple geometric dimensions, i.e. zigzag multi-persistence. However, such techniques are barely explored even in pure mathematics.
>
> **Q2**: The motivation for employing adaptive learning rates derived from topological features requires more explanation or stronger theoretical justification.
>
> **A2**: Thank you for your suggestion. Qualitatively speaking, persistent homology captures essential "shape" features across scales, directly reflecting graph structure. By shape here, we understand properties invariant under continuous transformations, e.g., bending, stretching, and twisting. When structural changes between snapshots are minimal, persistent homology features remain stable, allowing smaller learning rates as parameter updates stabilize. Conversely, significant structural changes produce substantial shifts in persistent homology features, necessitating larger learning rates to adapt.
>
> We designed an experiment to illustrate this point. Starting from a random BA network A with 500 nodes, we randomly changed m% of the edges between each snapshot 50 times. Experiments are run under both TMetaNet and fixed learning rate methods, where m takes values of 5, 20, 50, 90. The experimental results of MRR are:
>
> ||5%|20%|50%|90%|
> |-|-|-|-|-|
> |TMetaNet|4.53±0.24*|4.15±0.11|3.56±0.15*|2.64±0.10*|
> |fixed|3.90±0.37|3.82±0.16|2.51±0.22|1.74±0.06|
>
> The performance of the fixed learning rate method drops faster with increasing m, while TMetaNet's performance drops relatively less, indicating that TMetaNet can better adapt to the changes in the graph structure and maintain good performance when facing random perturbations of the graph structure.
>
> **Q3**: The authors adopted two settings to train the model and the results are clearly provided. According to Table 1, TMetaNet's performance under "WinGNN Setting" is significantly better than that under "Live Update Setting" consistently while the other methods' performance are not.
>
> **A3**: Our baseline performance is close to what WinGNN and ROLAND reported, which aligns with the observation that performance under live update setting is better than WinGNN. We believe this difference mainly stems from  variations in how our method is implemented under different settings. In the WinGNN setting, we configure TMetaNet's learning rate updates based on WinGNN's model without temporal encoders. Since WinGNN itself uses meta-learning to update node embeddings, TMetaNet's meta-learning rate updates can more significantly enhance WinGNN's performance, as shown in the table. This explains why TMetaNet performs better under the WinGNN setting compared to the live update setting.
>
> **Q4**:Are there any existing works that use PH for dynamic graphs?
>
> **A4**:Yes, there are several works that apply PH to dynamic graphs. For example, [1] designs a stable distance between dynamic graphs based on persistent homology; [2] uses neural networks to approximate Dowker persistent homology for dynamic graphs, and [3] applies PH in diffusion models for dynamic graphs. These papers, along with other related approaches, primarily focus on using PH to represent dynamic graphs for downstream tasks, while our work specifically focuses on using time-evolving PH to guide parameter updates in meta-learning models.  Furthermore, there are yet no studies on explicit integration of {\bf time-evolving topological information} for link prediction, either in a form of zigzag persistence or any other alternative approach.
>
> [1] Stable distance of persistent homology for dynamic graph comparison
> [2] Dynamic Neural Dowker Network: Approximating Persistent Homology in Dynamic Directed Graphs
> [3] Topological Zigzag Spaghetti for Diffusion-based Generation and Prediction on Graphs

---

### Official Review · Reviewer_K9Wc · 2025-03-13

**Overall Recommendation:** 3

**Summary:**

This paper proposes TMetaNet, a topological meta-learning framework for dynamic link prediction that integrates DZP to capture high-order topological features in dynamic graphs. The authors claim that DZP provides a computationally efficient and stable representation of dynamic graph evolution, which is then used to guide meta-learning parameter updates. Theoretical stability guarantees for DZP are provided, and ablation studies validate the necessity of topological features.

**Claims And Evidence:**

Claim 1: we propose the Dowker Zigzag Persistence (DZP), a computationally efficient and stable dynamic graph persistent homology representation method...
Evidence: The complexity analysis of DZP are mentioned in Section 4, however, it lacks direct comparisons with traditional Zigzag Persistence. Table 4 shows significant runtime increases for TMetaNet under the ROLAND and WINGNN settings, although the authors analyzed the reasons, which may limit practicality.

**Essential References Not Discussed:**

This work overlooks recent SOTA baselines on DTGB, such as:

[1] Yanping Zheng, Zhewei Wei, and Jiajun Liu. 2023. Decoupled Graph Neural Networks for Large Dynamic Graphs. Proc. VLDB Endow. 16, 9 (May 2023), 2239–2247. https://doi.org/10.14778/3598581.3598595

[2] Fu, J., Guo, X., Hou, J. et al. SEGODE: a structure-enhanced graph neural ordinary differential equation network model for temporal link prediction. Knowl Inf Syst 67, 1713–1740 (2025)

In addition, some past work combining topology and dynamic graphs can be included in related work, such as:
[3] Zhou, Zhengyang et al. “GReTo: Remedying dynamic graph topology-task discordance via target homophily.” International Conference on Learning Representations (2023).

**Experimental Designs Or Analyses:**

Cross-dataset experiments are comprehensive. The improvements on link prediction performance are visible under most datasets. In addition, the ablation study and noise analysis experiment are relatively complete.

**Methods And Evaluation Criteria:**

Metrics (ACC, MRR) are appropriate for the dynamic link prediction task.

**Other Comments Or Suggestions:**

1.	Discuss the scalability of TMetaNet on large-scale graphs.
2.	Discuss the possible solution of reconstructing TMetaNet to deal with continuous-time dynamic graphs (CTDGs) since CTDGs are more consistent with real dynamic graph scenes and more informative.

**Other Strengths And Weaknesses:**

Strengths:
1.	The paper is well-written with comprehensive experiments covering LP tasks, Ab study, noise robustness and hyperparameter sensitivity analysis.
2.	The integration of Dowker complexes with Zigzag Persistence is relatively novel.

Weaknesses:
1.	The model has high computational overhead (as shown in Table 4), which limits the scalability.
2.	There is insufficient explanations for certain results (e.g., Reddit-Body performance drop in Table 1). The authors are expected to further explain the performance drop.
3.	The idea of introducing graph structure topological information to graph learning to improve model performance is not a new idea for both static and dynamic graphs. Hence, the starting point of this work is a relatively incremental motivation.

**Questions For Authors:**

See above weaknesses and comments.

**Relation To Broader Scientific Literature:**

This work is related to the research of topology.

**Theoretical Claims:**

Theorem 4.5 (DZP stability) relies on discrete ϵ-smoothing and tripod constructions in Appendix B. However, critical lemmas (e.g., Lemma B.3) lack rigor, for instance, the composite tripod construction and its temporal consistency are not fully justified.

---

> ### Author Rebuttal · Authors · 2025-04-01
>
> **Q1**: Lemma B.3.
>
> **A**: To clarify briefly: given two tripods
> $R_1:\mathcal{G}^X \leftarrow W_1 \rightarrow \mathcal{G}^Y$ and $R_2:\mathcal{G}^Y \leftarrow W_2 \rightarrow \mathcal{G}^Z$ , each satisfying temporal consistency, their composite tripod is defined via fiber product: $W=\{(w_1,w_2)\in W_1\times W_2\mid\pi_2^{(1)}(w_1)=\pi_1^{(2)}(w_2)\}.$ Temporal consistency holds because nodes align through intermediate set $V_t^Y$: $(\pi_1^R)^{-1}(V_t^X)\leftrightarrow(\pi_2^{(1)})^{-1}(V_t^Y)=(\pi_1^{(2)})^{-1}(V_t^Y)\leftrightarrow(\pi_2^R)^{-1}(V_t^Z)$
>
> **Q2**: Task-splitting of Fig.5.
>
> **A**:  ROLAND splits each snapshot $G_t$ into train/val/test sets. The model trains on $G_{t-1}$'s training set, validates on $G_t$'s validation set, and tests on $G_t$'s test set, utilizing all snapshots for both training and testing. WinGNN uses chronological splitting, dividing the sequence into training and testing periods. E.g., in a 6-snapshot sequence, WinGNN uses the first 4 snapshots for training and the last 2 for testing.
>
> **Q3**:  Recent SOTA baselines.
>
> **A**:  We have run DeGNN [1] as a baseline (∗ indicates statistical significance). We found that TMetaNet outperforms DeGNN in both settings. We are currently implementing [2] and will post the results later. The topology [3] refers to the specific connection states between nodes in a dynamic graph and their changes over time. In our paper, we use term topology in terms of algebraic and computational topology on graphs, i.e., shape characteristics of various orders which provide important information about higher-order structural organization of the graph.
>
> ROLAND setting
> ||ALPHA|OTC|BODY|TITLE|UCI|ETH|
> |-|-|-|-|-|-|-|
> |DeGNN|ACC|76.7±1.33|76.9±1.33|89.7±3.6|92.6±3.5|76.4±1.12|62.6±1.73|
> ||MRR|12.5±1.03|15.0±1.21|26.7±3.5|40.1±4.8|9.2±4.3|33.0±2.62|
> |TMetaNet|ACC|86.84±1.02*|85.89±1.22*|89.59±1.17|93.96±0.02*|80.88±0.08*|85.10±1.46*|
> ||MRR|17.68±0.55*|18.06±1.22*|34.93±1.07*|42.72±1.01*|10.99±0.92*|38.08±1.57*|
>
> WinGNN setting
> ||ALPHA|OTC|BODY|TITLE|UCI|ETH|
> |-|-|-|-|-|-|-|
> |DeGNN|ACC|81.48±2.87|81.87±0.08|OOM|OOM|75.11±1.02|OOM|
> ||MRR|32.36±0.90|29.85±0.59|OOM|OOM|20.15±1.13|OOM|
> |TMetaNet|ACC|89.92±1.84*|90.43±1.17*|98.26±1.29*|99.63±0.07*|86.37±5.63*|97.83±1.53*|
> ||MRR|38.93±3.06*|39.98±2.16*|28.93±2.06*|34.96±2.06*|25.31±1.02*|78.07±1.09*|
>
> **Q4**: Scalability of TMetaNet.
>
> **A**: On ALPHA data, compared to VR complexes Zigzag Persistence, our method reduces the computational overhead by 46% on average per snapshot during complex construction. When dealing with extremely large-scale graphs, we can sample from snapshots or remove nodes according to degree centrality from low to high to obtain subgraphs that preserve global higher-order features, and then calculate the learning rates.
>
> **Q5**: Insufficient explanations.
>
> **A**: For Reddit-Body data, the differences between adjacent snapshots are relatively small, leading to suboptimal performance under the ROLAND setting, as the complementary information gains yielded by our method are less profound. However, under the WinGNN settings where we aggregate training within a certain window length, extracted topological signals are more prominent; TMetaNet more efficiently leverages the underlying higher-order graph topology, and outperforms the baseline. These findings indicate that extracted time-evolving topological information and the associated induced learning rates have higher value under more heterogeneous dynamics and less value for more homogeneous cases.
>
> **Q6**: The starting point.
>
> **A**: Topological information has been indeed incorporated into graph learning tasks before, largely for static scenarios and most recently for dynamic scenarios. However, there are no studies yet on explicit integration of time-evolving topological information for dynamic link prediction. This is also the first work that specifically focuses on using time-evolving topological representation to guide parameter updates in meta-learning models. Finally, models for dynamic link prediction and graph meta-learning using even conventional topological information are yet in their nascency. Our method makes a step forward toward this important direction by explicitly incorporating the essential time-evolving topological information in a mathematically rigorous and computationally more efficient manner.
>
> **Q7**: CTDGs.
>
> **A**: Indeed, CTDGs provide a more fine-grained node evolution process through event streams, by using representations via ODEs. However, CTDGs primarily focus on pairwise node interactions and do not account for simultaneous multi-node structures. TMetaNet and CTDGs can effectively complement each other. One approach is to use PDE rather than ODE in CTDG, where PDE describes dependencies across all nodes in the neighborhood and TMetaNet is used to parametrize the resulting coupled dynamics. Another direction is to use CTDG for identification and reconstruction of the dynamic landmarks in TMetaNet.

---

### Decision · Program_Chairs · 2025-05-01

**Decision:**

Accept (poster)

**Comment:**

The paper presents a topological meta-learning framework for dynamic link prediction. All reviewers agree that the approach is novel and claims are largely backed up by experiments. From my perspective, the (extensive) author rebuttal also addresses the main concerns, which is even appreciated in (some) reviewer's response(s); I am recommending (weak) acceptance at this point, but encourage the authors (1) to include any additional experiments in the final version and (2) to include the rebuttal comments as best as possible.